# Non-Invasive Monitoring of Vital Signs for the Elderly Using Low-Cost Wireless Sensor Networks: Exploring the Impact on Sleep and Home Security

Carolina Del-Valle-Soto [1,*,†], Ramon A. Briseño [2,†], Leonardo J. Valdivia [1], Ramiro Velázquez [3] and Juan Arturo Nolazco-Flores [4]

1   Facultad de Ingeniería, Universidad Panamericana, Álvaro del Portillo 49, Zapopan 45010, Mexico; lvaldivia@up.edu.mx
2   Centro Universitario de Ciencias Económico Administrativas, Universidad de Guadalajara, Zapopan 45180, Mexico; alejandro.bmartinez@alumnos.udg.mx
3   Facultad de Ingeniería, Universidad Panamericana, Aguascalientes 20296, Mexico; rvelazquez@up.edu.mx
4   School of Engineering and Science, Tecnólogico de Monterrey, Monterrey 64849, Mexico; jnolazco@tec.mx
*   Correspondence: cvalle@up.edu.mx; Tel.: +52-33-1368-2200
†   These authors contributed equally to this work.

**Abstract:** Wireless sensor networks (WSN) are useful in medicine for monitoring the vital signs of elderly patients. These sensors allow for remote monitoring of a patient's state of health, making it easier for elderly patients, and allowing to avoid or at least to extend the interval between visits to specialized health centers. The proposed system is a low-cost WSN deployed at the elderly patient's home, monitoring the main areas of the house and sending daily recommendations to the patient. This study measures the impact of the proposed sensor network on nine vital sign metrics based on a person's sleep patterns. These metrics were taken from 30 adults over a period of four weeks, the first two weeks without the sensor system while the remaining two weeks with continuous monitoring of the patients, providing security for their homes and a perception of well-being. This work aims to identify relationships between parameters impacted by the sensor system and predictive trends about the level of improvement in vital sign metrics. Moreover, this work focuses on adapting a reactive algorithm for energy and performance optimization for the sensor monitoring system. Results show that sleep metrics improved statistically based on the recommendations for use of the sensor network; the elderly adults slept more and more continuously, and the higher their heart rate, respiratory rate, and temperature, the greater the likelihood of the impact of the network on the sleep metrics. The proposed energy-saving algorithm for the WSN succeeded in reducing energy consumption and improving resilience of the network.

**Keywords:** algorithm; sensors for healthcare; wireless sensor networks (WSN); energy-saving algorithm

## 1. Introduction

Research on Wireless Sensor Networks (WSN) has been conducted across different fields, such as in agronomy, the military field, livestock, poultry, home automation, and vehicular traffic [1]. Their objective is to monitor the phenomena that occur, determine or take actions depending on the results obtained and extract useful information to reprogram systems through increasingly intelligent algorithms.

WSN and communication protocols are useful in medical applications to develop monitoring systems and apply the benefits of technology focused on assisting elderly patients. Improved healthcare monitoring becomes possible if it can take values from either a phenomenon or object that is in motion or static to integrate Information and Communications Technology (ICT) with medicine to monitor a patient's principal vital signs and investigate

the operation of wireless transmission. This work focuses on monitoring and applying the system for older adults, the most vulnerable population in society [2]. Improved monitoring of older adults could reduce mortality. The proposed sensor system is meant to be used in the comfort of the patient's home and provides safety recommendations on a daily basis.

Sensor networks and their performance are increasingly being studied in health-monitoring applications. They greatly help the medical field by obtaining online, real, and highly available data [3]. A large number of tests carried out with the help of such sensors have already been conducted, from monitoring body temperature to smart sensors that help tissue reconstruction or surgical interventions [4,5]. In medicine, various values to be sensed from the human body are considered critical when evaluating a patient. These data must be considered for any diagnosis, becoming proactive monitoring where the physician can observe the values generated online without having to be present in person [6]. In addition, there are fundamental aspects when diagnosing a patient that must be considered. These vital signs are considered essential when evaluating a patient's health status since they can provide an accurate signal of human body function before giving a diagnosis based on experience, symptoms, and tests carried out [7].

In older adults, due to their advanced age, it is difficult for them to make periodic visits to specialized health centers. This consequently gives rise to the need to adequately monitor the principal vital signs. This makes it possible to monitor them more effectively, avoid health status complications, and collect statistics of their different values [8]. Continuous monitoring of the state of health guarantees timely care by the doctor, preventing their health from deteriorating. Using sensors, this periodic monitoring is conducted remotely without the need for the patient to attend specialized health centers or for the doctor to have to mobilize. It is also helpful in maintaining personal reports with the possibility of sending them to relatives or loved ones and having an updated overview of the day-to-day health status of the older adult [9]. The study and application of WSN and the implementation of prototypes allow us to have a clearer idea of what is intended to be monitored. In this case, they would be the main vital signs of the elderly and some metrics related to sleep and the way we sleep. By having a system that allows this type of monitoring to be carried out, it is about positively impacting the quality of life of the elderly. With a timely assessment, future inconveniences can be avoided, premature death can be avoided, and appropriate actions can be taken.

Having knowledge on how people sleep is essential to determine their state of health, especially when they suffer from stress or diseases such as Alzheimer's, Parkinson's, or other conditions that affect rest [10]. To monitor sleep, sensors and electrodes are needed, which are uncomfortable and, in turn, can further impair mental rest. The technological opportunity is great because there is the need to better understand sleep, and a large fraction of the population needs help in sleeping [11]. REM and non-REM sleep alternate in sleep cycles, which last about 90 min. REM sleep is characterized by rapid random eye movement, dystonia, and vivid dreaming. It is also known as paradoxical sleep because of the physiological similarities to waking states, including rapid, and low-voltage desynchronized brain waves. REM sleep produces marked physical changes, including the suspension of central homeostasis, which allows for large fluctuations in respiration, thermoregulation, and circulation that do not occur in any other mode of sleep or wakefulness [12].

The main contributions of this research work lie in the development and application of a wireless sensor network (WSN) for monitoring the vital signs of elderly individuals in their homes to improve their sleep quality. The environment is the same as the home of the elderly. The sensor network is installed in their home to be as strange or invasive as possible. In this study, we use WSNs with low-cost sensors, including motion, pressure, temperature, humidity, noise, light, gyroscope, and air quality sensors, to gather real-time data from different house areas. The objective is to analyze the sleep metrics of older adults before and after implementing the sensor network to determine the impact of the system on improving rest. The proposed system aims to provide daily safety recommendations to the elderly and enable remote monitoring of vital signs, helping them to live independently

and receive timely medical care. This work addresses the challenges of monitoring elderly individuals, especially in terms of their sleep patterns, as this age group tends to have reduced mobility and difficulty accessing health centers. We demonstrate the potential of using WSNs and intelligent algorithms to optimize sensor utilization, improve sleep quality, and provide personalized recommendations for the elderly's well-being. The sensor system has a group of pre-written recommendations based on the possible causes of sensor parameter anomalies. These anomalies may be due to the person sleeping in a noisy environment, getting up several times during the night, or uncomfortably cold or warm temperatures in their home. They are only approximations of possible causes to improve the quality of life of people who live alone. This, being an approximation, represents one of the system's limitations. The findings of this study highlight the importance of continuous monitoring and non-invasive technology in enhancing the quality of life for older adults and preventing health complications. Overall, this work contributes to advancing remote monitoring systems for elderly individuals, particularly in the context of sleep monitoring and healthcare assistance.

*Motivation*

This work aims to find relationships between parameters impacted by the sensor system and predictive trends based on the threshold of measurements perceived by the sensors during the day. This proposed system consists of a low-cost WSN deployed at the home of seniors who want to remain independent. In the context of the provided passage, an "independent person" refers to an elderly individual who wishes to live independently and maintain their autonomy without relying heavily on constant assistance or care from others. This could be an elderly person who prefers to live in their own home rather than moving to an assisted living facility or nursing home.

This system monitors the main areas of the house programmed with a low-energy cost algorithm and sends recommendations to the person at the beginning of each day. It is a simple algorithm based on the threshold of measurements perceived by the sensors during the day. In this way, the nodes change hierarchies in the network and has proactive or reactive priorities for sending information. The routing protocol takes advantage of each node's proactive or reactive nature to transmit the packets. The change that is optimized in the proposed algorithm is that when the node's measurement levels are kept within a normal range, the protocol responds to sending packets reactively. Meanwhile, if the measurement levels are outside the threshold, the protocol responds proactively to sending packets to better control the information review.

The main contribution of this work is the novel categorization of the impact of metrics related to the person's vital signs based on the way the person sleeps. By analyzing the sleep metrics of older adults before and after using the proposed sensor network, we will be able to determine the level of impact (high, medium, or low) that the system will have on improving rest once we know the person's metrics prior to installing the sensor network. Therefore, we discuss the usability of our proposed system from the perspective of a non-invasive technological product and its use as a medical tool to improve people's rest.

We found that recommendations based on measures from the sensors improved sleep quality. In addition, the proposed energy-saving algorithm assists in optimizing the network's performance and directing its efforts towards areas that can have the greatest impact on improving sleep quality for individuals.

The remainder of this paper is organized into five sections. Below is the Related Work Section, in which the comparative relationship of the state of the art is explained. The Materials and Methods section describes the presentation of the sensor network, the routing algorithms, and the correlation metrics. The Results section explains the impact and relationship of the parameters measured in people and their relationship with vital signs to improve sleep quality. The results are also discussed here. Finally, in the Conclusions section, the improvement of the proposed system and possible areas of opportunity it still presents are given.

## 2. Related Work

Remote monitoring systems for seniors in homes refer to technology-based systems that allow for the monitoring of elderly individuals at their own homes, without the need for in-person assistance. These systems often use sensors and wearable devices to collect data about the individual's health, activity levels, and environment.

The state of the art in remote monitoring systems with sensor networks in smart homes for the elderly is an active field of research, with many studies and projects being conducted to improve the capabilities and effectiveness of these systems. Some recent related works include the use of wearable sensors to monitor vital signs such as heart rate and oxygen saturation, as well as the use of cameras and other non-wearable sensors to track the individual's movements and activities. Researchers are also exploring the use of machine learning and artificial intelligence techniques to analyze the data collected by the sensors and detect potential issues or changes in the individual's health or well-being. Other related works include the use of virtual assistants and other forms of human–computer interaction to provide support and assistance to the elderly individuals, as well as the use of blockchain technology to ensure the security and privacy of the data collected by the system.

The exploitation of technology and, especially, sensor-based information networks, is considered an effective solution for social assistance. With the advantages provided by technology, several studies [13–15], have succeeded in implementing embedded systems to achieve constant monitoring of human body values.

Specialists and researchers from different sciences, health professionals, and scientists devoted to study biomedicine are currently developing embedded systems to monitor patients' health in search of real data. The works cited in [16,17] conducted studies based on technological tools to work in unusual places without human beings and with equipment with minimal energy consumption to process, transmit, and visualize information.

Implementing and fine-tuning remote monitoring and assistance systems is a highly complex process that is attempting to be solved by building smart homes. Researchers of the work cited in [18] designed a device based on an artificial intelligence algorithm that analyzes the signals around the person to assess their level of sleep in light, deep, or REM. Other studies [19], reveal that low-power radio waves that detect small changes in body movement caused by the patient's breathing and pulse rate can non-intrusively diagnose and study sleep problems.

Studies [20] have shown that remote monitoring systems can positively impact sleep practices in seniors. For example, wearable devices that track sleep patterns can provide useful insights into the individual's sleep quality and habits, allowing for the identification of any sleep-related issues and the implementation of interventions to improve sleep. Additionally, remote monitoring systems [21] can also provide a sense of safety and security, which can help to reduce anxiety and promote better sleep.

However, there are also concerns that the constant monitoring and use of technology can have a negative impact on sleep practices [22]. For instance, exposure to screens and other sources of blue light before bedtime can disrupt the natural sleep–wake cycle and lead to insomnia. Furthermore, the continuous use of technology can also increase the likelihood of being woken up during the night by notifications or alarms from the monitoring system. Remote monitoring systems for seniors in homes can have both positive and negative impacts on sleep practices [23]. It is important to consider the potential benefits and drawbacks of these systems when deciding on the best approach for monitoring elderly individuals at their own homes, as in a previews work cited in [24].

Table 1 describes a comparison of metrics with related work addressing sensors and/or wearables for sleep monitoring. Remote home monitoring systems for seniors are designed to allow seniors to live independently while providing support and assistance as needed. These systems typically include a variety of sensors and wearables that can track various aspects of the senior's health and well-being, including sleep patterns. Some common metrics that are used in these systems include heart rate, oxygen levels, and movement patterns. Other metrics may include sleep duration, sleep quality, and number

of awakenings. The system configuration management in these systems may include the use of wireless sensors, mobile devices, and cloud-based services to monitor and collect data. The application domain of these systems is typically in the field of healthcare and aging, with the goal of providing seniors with the support they need to live independently for as long as possible.

**Table 1.** Comparison of metrics with related work addressing sensors and/or wearables for sleep monitoring.

| Reference | Metrics | System Configuration Management | Application Domain | Brief Description |
|---|---|---|---|---|
| [25] | Sleep duration, sleep efficiency, movement detection | Wireless sensor network | Elderly care | A WSN system that uses motion sensors to monitor sleep patterns and movement of seniors in their homes, with the aim of promoting independent living. |
| [26] | Sleep stages, heart rate variability, respiratory rate | Wearable device | Sleep medicine | A wearable device that utilizes various sensors to measure physiological parameters and classify sleep stages, with the aim of improving diagnosis and treatment of sleep disorders. |
| [21] | Sleep duration, activity levels, ambient light, temperature | Smart home system | Geriatric care | A smart home system that uses a combination of sensors and cameras to monitor sleep and activity patterns of seniors, with the aim of detecting abnormal behavior and providing health interventions. |
| [27] | Sleep efficiency | Wireless sensor network | Elderly population | A WSN for monitoring sleep efficiency in the elderly population. |
| [28] | Sleep duration | Smartwatch | Healthy adults | A smartwatch-based system for measuring sleep duration in healthy adults. |
| [29] | Sleep quality | Body-worn sensors | Athletes | A body-worn sensor system for assessing sleep quality in athletes. |
| [30] | Sleep efficiency, Sleep duration, Sleep fragmentation, Sleep quality | Smartphone-based application | General population | A smartphone-based application for monitoring sleep efficiency, sleep duration, sleep fragmentation, and sleep quality in the general population. |
| [31] | Sleep efficiency, sleep latency, wake after sleep onset, total sleep time | Wearable actigraphy device + remote monitoring system | Independent living for older adults | A feasibility study of using wearable actigraphy devices and a remote monitoring system to track sleep patterns in older adults living independently. |
| [32] | Sleep efficiency, sleep latency, wake after sleep onset, total sleep time | Wearable actigraphy device + remote monitoring system | Independent living for older adults | A randomized controlled trial of using wearable actigraphy devices and a remote monitoring system to track sleep patterns in older adults living independently. |
| [33] | Sleep efficiency, sleep latency, wake after sleep onset, total sleep time | Wearable actigraphy device + remote monitoring system | Independent living for older adults | A pilot study of using wearable actigraphy devices and a remote monitoring system to track sleep patterns in older adults living independently in a community setting. |

**Table 1.** *Cont.*

| Reference | Metrics | System Configuration Management | Application Domain | Brief Description |
|---|---|---|---|---|
| [34] | Sleep efficiency, sleep latency, wake after sleep onset, total sleep time | Wearable actigraphy device + polysomnography | Independent living for older adults | A comparison of using wearable actigraphy devices and polysomnography to track sleep patterns in older adults living independently. |
| [35] | Sleep efficiency, sleep latency, wake after sleep onset, total sleep time | Non-contact sleep monitoring system | Independent living for older adults | An assessment of using a non-contact sleep monitoring system to track sleep patterns in older adults living independently. |

Table 2 provides a comprehensive comparison of various sleep monitoring mechanisms, including proposals from the literature and the state of the art. Each mechanism is evaluated based on its unique features and contributions. These existing mechanisms offer valuable insights into sleep monitoring, such as detecting sleep stages, assessing stress levels, monitoring sleep apnea, analyzing sleep patterns, observing sleep behavior, and tracking snoring. However, our proposed system, based on a non-invasive wireless sensor network (WSN) located in people's homes, introduces significant advancements in sleep monitoring technology. It operates without the need for wearables, making it more comfortable for users. The system monitors various parameters, such as luminosity, noise, and movement, providing a comprehensive view of the sleep environment and body parameters. By integrating multiple parameters, the proposed system can offer more accurate and personalized insights into sleep quality, generating personalized sleep recommendations. This non-wearable WSN in homes ensures a seamless and convenient monitoring experience, offering valuable improvements over existing mechanisms. Overall, the proposed system represents a promising and innovative approach to sleep monitoring, enhancing user comfort and providing actionable recommendations for better sleep quality.

**Table 2.** Comparison of Sleep Monitoring Mechanisms.

| Mechanism | Differences | Improvements | System Type |
|---|---|---|---|
| Roll-over detection [36] | Measures body movement. | Detects sleep stages. | Wearable |
| Wearable heart rate sensor [37] | Tracks heart rate. | Assesses stress levels. | Wearable |
| Real-time wearable system [38] | Measures blood oxygen levels. | Monitors sleep apnea. | Wearable |
| IoT-assisted wearable sensor [39] | Records brain activity. | Analyzes sleep patterns. | Wearable |
| Non-wearable IoT-based smart ambient [40] | Captures visual information. | Observes sleep behavior. | Non-wearable |
| Internet of Intelligent Things [41] | Records ambient sounds. | Monitors snoring. | Non-wearable |
| **Our proposed system (Non-invasive WSN)** | • Monitors luminosity, noise, movement, etc. <br> • Generates personalized sleep recommendations. | • Non-invasive, no need for wearables. <br> • Comprehensive monitoring of sleep environment and body parameters. <br> • Personalized recommendations for better sleep quality. | **Non-wearable WSN in Homes** |

Table 3 provides a comprehensive comparison of various state-of-the-art algorithms for energy-saving in remote monitoring of older adults at home. It highlights the different types of algorithms, their impacting metrics, and the percentage of global consumption reduction they can achieve. The algorithms are classified based on their type, which includes sleep-based, tree-based, collaborative, multi-objective, cluster-based, threshold-based, and reactive mode based. The impacting metrics of the algorithms include energy efficiency, network lifetime, latency, QoS, and routing efficiency. The percentage of global consumption reduction ranges from 15% to 60%. Overall, the table aims to provide insights into the different algorithms that can be used to enhance energy efficiency in remote monitoring systems, and their effectiveness in achieving energy consumption reduction.

**Table 3.** Comparison of State-of-the-Art Energy Saving Algorithms for Remote Monitoring of Older Adults at Home.

| Algorithm | Type | Impacting Metrics | Global Consumption Reduction |
|---|---|---|---|
| Sleep-Scheduler [42] | Sleep-based | Energy efficiency, latency | 40–60% |
| Energy-Aware Tree-Based Routing [43] | Tree-based | Network lifetime, energy efficiency | 15–40% |
| Distributed Collaborative Routing [44] | Collaborative | Energy efficiency, routing efficiency | 50% |
| Multi-Objective Optimized Routing [45] | Multi-objective | Energy efficiency, network lifetime | 30–50% |
| Adaptive Clustering with QoS [46] | Cluster-based | Energy efficiency, QoS, network lifetime | 30–50% |
| Adaptive compression [47] | Threshold-based | Energy efficiency, latency | 50–60% |
| QoI-aware [48] | Sleep-based | Energy efficiency, latency | 50% |
| Our work | Reactive mode based | Energy efficiency, latency | 50% |

## 3. Materials and Methods

This section presents the monitoring system of sensors, routers, and concentrators with its technical specifications. In addition, real photos of some installations of the nodes and their power supply are shown. The technical description of each type of system device is provided. Finally, we present the methodology of the experiment and the analyses carried out on the information of the vital signs metrics measured for each person during the experiments.

The connection between sensors, security, and the activities of the elderly population is multifaceted:

Sensor System for Monitoring: The sensor system collects data from various sensors, including motion, pressure, temperature, humidity, noise, light, gyroscope, and air quality sensors. These sensors are designed to monitor the home environment and the elderly individual's vital signs while being unobtrusive and comfortable for the user.

Security and Privacy: The work emphasizes the need for non-invasiveness and respect for the privacy of the elderly. The data collected by the sensor network must be securely transmitted and stored to protect the individual's sensitive health information. Ensuring data security and privacy is crucial in healthcare applications, especially when dealing with sensitive data from vulnerable populations.

Activities of the Elderly: The sensor system aims to capture data related to the activities of the elderly in their homes. It tracks sleep patterns, movement, noise levels, and air quality to identify anomalies and provide personalized recommendations for improving sleep quality. By understanding the person's daily activities, the system can offer relevant and targeted suggestions to enhance their well-being.

Wireless Sensor Network (WSN) and Communication Protocols: The work highlights the importance of using WSNs and communication protocols to efficiently collect and transmit data from the sensors. The system uses wireless communication to transmit the collected data to a central hub or concentrator node, where it is processed and analyzed. The use of efficient communication protocols is crucial for optimizing data transmission and reducing energy consumption.

Energy-Saving Algorithm: The proposed energy-saving algorithm is integrated into the sensor system to optimize power consumption. It allows the system to be proactive or reactive, depending on the level of change in the measured parameters. Proactive mode ensures continuous monitoring when parameters show high variability, while reactive mode conserves energy by collecting data only on demand when the parameters are stable.

The impact of this work lies in its potential to revolutionize remote monitoring systems for elderly individuals, particularly in the context of sleep monitoring and healthcare assistance. By using WSNs and intelligent algorithms, the system can efficiently collect and analyze data, providing real-time feedback and personalized recommendations to improve sleep quality and overall health. The non-invasive nature of the system allows elderly

individuals to receive care and assistance without the need for constant visits to healthcare centers, thus improving their quality of life and reducing the risk of health complications.

### 3.1. System of Sensors Network

Table 4 shows the technical specifications of the proposed system's sensors encompassing 14 nodes. These sensors monitor the main areas of the house in order to know behaviors and be able to send alerts or notifications to the user via email or SMS. The implemented sensors are Motion, Pressure, temperature and humidity (PTH), Noise, Light, Gyroscope, and Air quality.

**Table 4.** System Sensor Specifications.

| Sensor | Description |
|---|---|
| Motion | DC 4.5–20 V, 50 µA delay: 5–200 S (adjustable) the range is (0.xx second to tens of second), Operation Temp.: −15–+70 degrees, Detection Range: 3 m to 7 m. |
| Pressure, temperature and humidity | Combines thermometer, barometer and hygrometer. Temperature range from −40 to +85 °C with an accuracy of ±1 °C and resolution of 0.01 °C, and for pressure 300–1100 hPa, accuracy of ±1 Pa, and resolution of 0.18 Pa. Supply voltage range: 1.71 V to 3.6 V. Accuracy tolerance ±3% relative humidity. |
| Noise | ULTRASONIC SENSOR HRXL-MAXSONAR. MAX4466 with adjustable gain. 20–20 KHz electric microphone. 2.4–5 VDC. 3.7 W. Frequency: 42 kHz. Type: Transmitter, Receiver. Maximum detection distance: 765 cm. Consumption: 2.1 mA. Operating temperature: −40 °C 65 °C. |
| Light | LDR photoresistor sensor module. Main chip: LM393. Minimum supply voltage: 3.3 V. Maximum supply voltage: 5 V. Output Type: Digital. Maximum rating: up to 38 V. |
| Gyroscope | 7A994. Axis Type: Single. Typical Angular Velocity (°/s): ±300. Typical Operating Supply Voltage (V): 3.3 | 5. Minimum Operating Temperature (°C): −40. Maximum Operating Temperature (°C): 105. Linearity: No. |
| Air quality | ZPHS01C Multi-in-One Air quality monitoring Sensor Module. Target Gas:PM2.5, CO2, CH2O, TVOC, Temperature and Humidity. Applications: Gas detector, Air conditioner, Air quality monitoring, Air purifier, HVAC system, Smart home. |

Figure 1 shows real photos of the devices installed for the monitoring system. The distributed sensor network consists of a hub node, which is the device connected to the computer via a USB connection and receives and manages all the information received and transmitted. It has three router nodes with sensors connected to their communication ports, such as temperature, pressure, humidity, light, sound, and the others we have activated. The sensed parameters are sent via radio frequency to the network up to the concentrator node, which can act as a repeater. The power supply can work in two types: (1) Input Voltage 1 (BAT): Acid/Lead Battery (12 V DC at 3.3 Ah). (2) Input Voltage 2 (CS): Solar Cell (12 V DC, 10 W, 0.5A). The graphic software of the network shows the logical neighbors of the node. It presents its relationship with each of its neighbors with respect to the RSSI (Received Signal Strength Indicator) and LQI (Link Quality Indicator) signals.

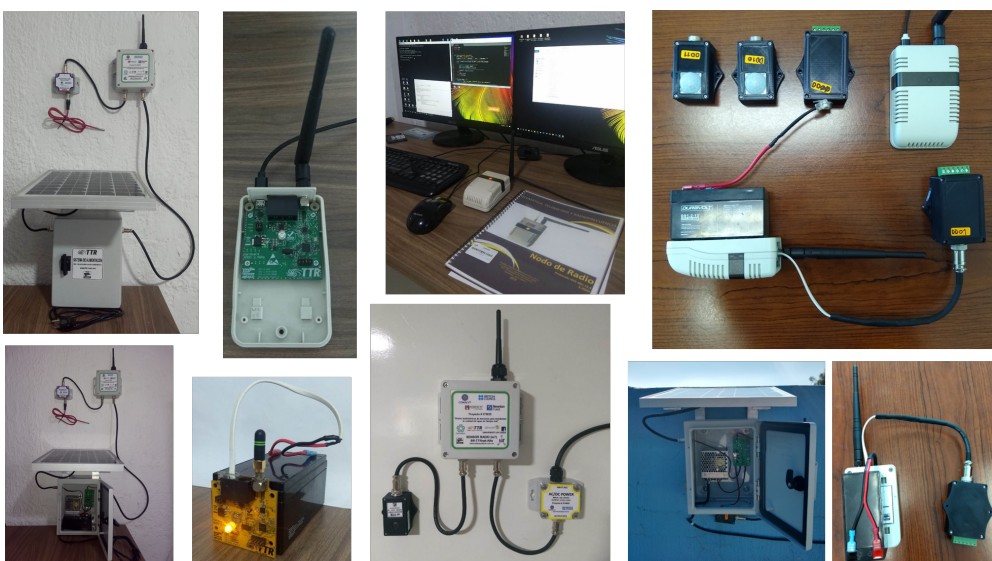

**Figure 1.** Real implementation of the sensor network.

Table 5 shows the main vital sign parameters related to sleep and how a person sleeps. A smartwatch can measure the person's main vital signs while he/she sleep. We show typical values for a person in a resting state who does not present cardiovascular or respiratory diseases. Upon the use of accelerometers and gyroscopes, we can monitor the most subtle movements. Pointing at the wrist, we have an optical sensor to capture the changes of each pulsation and calculate the oxygen level in the blood. When the heart rate is low, and there is hardly any movement, it is recorded as deep sleep. When there is much movement, and the heart rate is high, it is considered light sleep. To know the REM phase of sleep, data from heart rate and movement sensors are added to the respiration frequency to make the estimate. Also, we can know when the person is awake because they move enough to do so. Stress measurements can be approximated by measuring changes in the duration between beats, also known as HRV (Heart rate variability). HRV offers a non-invasive way to pinpoint imbalances in the autonomic nervous system. Based on data collected from many people, the variation between beats tends to be less if the system is in a fight or flight mode. In contrast, if the nervous system is relaxed, the variation may be more significant [49]. It is essential to mention that the consent notice given to each person notes that consuming caffeine, nicotine, alcohol, or drugs may alter the results. In addition, the measurements will not be very reliable in people with heart disease or asthma, who are doing sports, or who need to wear the watch correctly. For this last reason, these types of people have been excluded from the experiments.

The sensor network is trained with the person's routines and keeps sensors active in the areas where the person spends the most time. In addition, it recognizes patterns of open windows or doors within the person's routine so as not to send unnecessary alerts. The sensor system monitors noise, movement, air quality, level, the door turns, and lights to send notifications of recommendations for improving sleep quality monitored by a conventional smartwatch. The system has an alert algorithm that can improve the quality of the person's sleep. For example, recommendations for fluid consumption if the person gets up many times to go to the bathroom, decrease the amount of light in the sleeping area, close or open doors or windows according to the amount of noise, etc. Each day, the person receives recommendation messages based on the night immediately before via email or SMS. This can lead to a gradual improvement in sleeping habits. The sensor system is not invasive, and the person does not have to configure it constantly. The system has an intelligent energy algorithm incorporated according to the adaptation of the sensors and its continuous use in the different areas of the house.

**Table 5.** Vital sign metrics [50].

| Metric | Typical Values |
|:---:|:---:|
| Heart rate (HR) | 60 to 100 beats per min |
| Breathing frequency (BF) | 12 to 18 breaths per min |
| Temperature (T) | 97.8 °F to 99.1 °F (36.5 °C to 37.3 °C) |
| Oxygen saturation (OS) | 95–100% |
| Total sleep time (TST) | between 7 and 8 h |
| REM sleep (REMS) | between 20 and 25% of total sleep time |
| Deep sleep (DS) | between 10 and 20% of sleep time |
| Heart Rate Variability (HRV) | Higher HRV indicates a relaxed and less stressed state than a lower HRV |
| Snoring (S) | few or none of sleep time |

The algorithm described in Algorithm 1 is based on the operation of the sensor network in conjunction with the coordinator node. When the network is turned on, the coordinator sends packets to know the whole topology. Each node has a hierarchy, the measurement of its sensing parameter, RSSI value and LQI value (these last two related to the quality of the link). Each node takes the measurement of its parameter depending on the programmed *request-time*. The node differentiates its measurement based on the subtraction of the initial default value and the current value. If this difference value is greater than 50% of the default value of the steady-state measurement, the node increases its hierarchy. This means that the measurement is very changeable, and it is important to monitor the area constantly. So, if the node hierarchy is greater than 3, the LQI value is 50% greater than its typical value (in stable state), and the RSSI value is 50% greater than its typical value (in stable state), the node is activated in proactive mode. This means that it is essential to monitor the parameter every *request-time* because its value is highly variable. Otherwise, the node goes into reactive mode, which reduces its energy consumption and only requests the sensed value when necessary, that is, on demand. In this way, the network has an adaptive behavior according to the home of each person.

The node is the device that has or can have multiple sensors. Indeed, the RSSI and LQI values should not change in static nodes. However, this can be altered by other devices that the person activates one day or by environmental conditions. Furthermore, when these factors change, there are anomalous environmental conditions, precisely what we want to analyze to establish the recommendations.

Therefore, the sensor system is the same for anyone, but the sensors' operation, energy consumption, and recommendations will be personalized to each home. Table 6 provides recommendations for sleeping better at home, grouped into three different categories: Group 1, Group 2, and Group 3. Each group contains a list of seven different recommendations aimed at helping individuals improve their sleep quality. Group 1 recommendations focus on creating a comfortable and relaxing sleep environment. This includes closing the window to block out noise, using a comfortable mattress and pillows, using a noise machine or earplugs to block out background noise, keeping the room at a cool temperature, using a humidifier to improve air quality, keeping the room dark and using black-out curtains, and avoiding caffeine and heavy meals before bedtime. All these recommendations are aimed at creating a sleep-conducive environment that is comfortable and free from distractions. Group 2 recommendations focus on relaxation and preparation for sleep. These recommendations include practicing relaxation techniques before bedtime, keeping a consistent sleep schedule, using a weighted blanket, trying aromatherapy with essential oils, keeping electronics out of the bedroom, using a white noise machine, and trying a sleep mask. All these recommendations are aimed at relaxing the body and mind and preparing them for sleep. Group 3 recommendations focus on developing good sleep habits, and include establishing a bedtime routine, exercising regularly but not close to bedtime, avoiding blue

light exposure before bedtime, considering using a sleep aid, avoiding napping during the day, trying a natural sleep remedy like melatonin, making sure your bed is comfortable and supportive. All these recommendations are aimed at developing good sleep habits which are conducive for a good night's sleep.

---

**Algorithm 1** Algorithm pseudocode of the network system.

---

```
Start
all Nodes ON;
set request_time;
coordinator_node starts;

Per each node do:
hierarchy_i = 0;
set default_measurement_i;
set LQI_initial;
set RSSI_initial;

Per each request_time do:
    set measurement_i;
end
d_measurement_i = |default_measurement_i - measurement_i|;
if(d_measurement_i > (0.05 * default_measurement_i))
{
    hierarchy_i++;
}

if((hierarchy_i > 3) && (abs(LQI_initial - LQI_i) >
(0.05 * LQI_i)) && (abs(RSSI_initial - RSSI_i)
> (0.05 * RSSI_i))
{
    node_i under proactive mode;
}
else
{
    node_i under reactive mode;
}
if (hierarchy_i == 3)
{
    node_i sends recommendations(group 1);
}
else if(hierarchy_i >= 2)
{
    node_i sends recommendations(group 2);
}
    else if(hierarchy_i >= 1)
    {
        node_i sends recommendations(group 3);
    }
        else
        {
            no recommendations;
        }
end
```

---

**Table 6.** Recommendations for sleeping better at home.

| Group | Types of Recommendations |
|---|---|
| Group 1 | 1. Close windows 2. Use a comfortable mattress and pillows 3. Use a noise machine or earplugs 4. Keep the room at a cool temperature 5. Use a humidifier to improve air quality 6. Keep the room dark and use black-out curtains 7. Keep electronics out of the bedroom |
| Group 2 | 1. Practice relaxation techniques before bedtime 2. Keep a consistent sleep schedule 3. Use a weighted blanket 4. Try aromatherapy with essential oils 5. Avoid caffeine and heavy meals before bedtime 6. Use a white noise machine 7. Try a sleep mask |
| Group 3 | 1. Establish a bedtime routine 2. Exercise regularly but not close to bedtime 3. Avoid blue light exposure before bedtime 4. Consider using a sleep aid 5. Avoid napping during the day 6. Try a natural sleep remedy like melatonin 7. Make sure your bed is comfortable and supportive |

This Table contains recommendations adapted to the general functioning of an average house in Mexico in an urban area. Each one of the groups is related to characteristics of the house or of the people, which could directly or indirectly impact the way the person sleeps. For example, Group 1 presents recommendations for the house or its composition. Group 2 presents recommendations mainly related to the person, which can be part of their daily life and habits. Moreover, Group 3 has recommendations about the person's bedroom. These recommendations can be helpful for people who are seeking ways to improve their sleep quality, especially for those who are using remote assistance systems for monitoring their sleep patterns. These systems can track and monitor an individual's sleep patterns and provide feedback on areas where improvements can be made. By following these recommendations, individuals can improve their sleep quality and overall well-being.

*3.2. Methodology of the Experiment*

Regarding the implementation and comparison of the system to evaluate energy consumption with and without the proposed algorithm, we performed tests with 30 people during a period of four weeks. We analyzed the system in the first two weeks without implementing the energy-saving algorithm in the nodes. In this case, the nodes were working without reactivity based on parameter changes. For the next two weeks, we implemented reactivity in the nodes to observe the savings in energy consumption. In this way, we could test the algorithm's performance complemented with energy savings. It should be noted that the tests on the 30 people to analyze their vital signs and the impact on sleep were carried out with the complete algorithm, that is, with the proposed energy savings.

The work presented here advances the state of the art in proposing a sensor system for monitoring people at home and providing personalized recommendations for improving sleep quality based on anomalous levels of measurements. This system leverages sensor technology and data analysis to offer solutions to enhance sleep patterns. The development of a sensor system tailored for home monitoring is a significant step forward. This system likely consists of various types of sensors strategically placed within the living spaces to capture relevant data related to sleep patterns. These sensors could include ambient light sensors, temperature sensors, motion sensors, heart rate monitors, and even advanced devices like sleep-specific wearable technology. The work proposes a data analysis framework to process the data collected by the sensor system effectively. By analyzing and interpreting the data, the system can identify anomalous levels or patterns in a person's sleep-related measurements. Anomalies could include irregular sleep duration, abnormal heart rate variations during sleep, or disturbances in sleep cycles. The key innovation lies in the system's ability to provide personalized recommendations based on the anomalous measurements. These recommendations could range from changes in sleep habits, lifestyle adjustments, relaxation techniques, or even suggestions to consult a medical professional, depending on the severity of the anomaly detected. The energy-saving algorithm integrated into the sensor system is another crucial advancement. By utilizing the programming nature of the sensors (proactive and reactive), the system can efficiently manage power consumption and extend the overall lifespan of the sensors. This algorithm likely employs techniques like dynamic sampling, adaptive data transmission, or intelligent sleep/wake-up modes for the sensors, ensuring that they are only active when necessary.

This study is based on monitoring people's sleep through a smartwatch device. The experiment is carried out for two weeks without the sensor system and two other weeks with a sensor network installed in the main areas of the house of 30 elderly adults. Once the averaged metrics of the older adults for the two weeks without the use of the sensor network and with the use of the system are known, we proceed to the analysis of the information.

The age of the people ranges between 50 and 70 years. There are 16 women and 14 men. People are relatively healthy concerning their health. They do not have heart or respiratory diseases (because this could alter the results a bit). The collected data will likely

provide valuable insights into sleep patterns and overall well-being for this demographic. A consent form to approve the use of the system and its installation in their home is signed. The nodes adapt according to the layout of each person's house. There must be at least two nodes in the room where the person sleeps, and in the main areas of the house (kitchen, living room, and dining room), at least one node. The nodes are easy to install near a power outlet. The nodes' adaptability based on the layout of each person's home is crucial. This ensures that the sensor system can effectively capture relevant data points while being minimally invasive and seamlessly integrated into the participants' living spaces.

First, in order to demonstrate whether the variables changed or not in a statistically significant way with the use of the sensor network, hypothesis tests for paired samples were used. To define the relevant hypothesis test, whether it is a parametric *t*-test (see in Appendix A Equation (A1)) or a non-parametric Wilcoxon test (see in Appendix A Equation (A2)), the normality of the difference between each metric (with the sensor network—without the sensor network) is verified using graphical methods. If the graphical methods do not find reliability, the Kolmogorov–Smirnov hypothesis test is executed to verify normality for a sample size greater than 50, or the Shapiro–Wilk test for samples of size 50 or smaller (see in Appendix A Equations (A3) and (A4), respectively).

Second, we analyzed the correlations that the variables have with each other for the two scenarios separately (without the use of the sensor network and with the sensor network). To do this, the Pearson correlation coefficient is chosen (see in Appendix A Equation (A5)) for each pair of variables that are normally distributed, or a non-parametric correlation coefficient such as Spearman's (see in Appendix A Equation (A6)) for a pair of variables where at least one is not normally distributed. Therefore, the normality of the metrics is verified separately, and the appropriate correlation coefficient is used.

Third, we ran a K-means clustering algorithm on the metrics for the two scenarios (before and after using the sensor network) with the purpose of grouping people with similar performance in their sleep-related vital signs. In Appendix A, Equation (A7) shows the cluster's k-means algorithm.

To choose the best grouping number of clusters we used a couple of parameters, the higher silhouette score, and a min of 10 samples per cluster. The silhouette score is a coefficient that goes from −1 to 1, where a number close to 1 shows a better fitness between the elements of the clusters (see in Appendix A Equation (A8)).

Also, we decided to work with a min of 10 samples for cluster to make representative groups. Then, we identified which samples changed to a better performance cluster, to a worse performance cluster, and which ones remained in the same cluster after using the sensor network. This classification is important, as it was used to compare the metrics before and after utilizing the sensor network on sample groups with similar performance in their metrics. Furthermore, each sample was labeled based on its performance classification, and this label was used as the target variable in a multinomial multivariable logistic regression with a Logit parameter estimation (see in Appendix A.9) performed to describe the predictive power of each metric in odds ratios and the relationships among them on the potential impact of the sensor network.

## 4. Results

In this section, we describe the results of the sensor system and its operation from an energy perspective. Subsequently, we describe the results of the sensor system concerning metrics based on vital signs and how they impact the person's sleep mode.

### 4.1. Sensor Network Performance

The energy-saving algorithm implemented in the home monitoring sensor system exploits the proactive and reactive nature of the nodes. Such algorithm is based on the fact that if a node has more alerts due to significant measurement parameter changes, it will enter into a proactive monitoring operation. This benefits the network because the node is more vigilant. Every time, it will monitor the parameters because something may

not work correctly at home for the rest of the person. On the other hand, when a node is in reactive mode, it spends less power because its monitoring is based on need. This is fostered because the node does not present intermittent attention alerts, and the follow-up of its monitoring parameters is relatively average.

Based on Algorithm 1, the system presents an energy-saving concerning the regular operation of the wireless sensor network. For this result, we consider the system's use for one hour, and when we apply the algorithm, we observe that the use of the sensor batteries is reduced. This is only an added value to the system because its principal value is monitoring the conditions of the house to give recommendations based on sleep.

Table 7 compares the energy consumption of two different system schemes. The first row of the table shows the energy for the system without the energy consumption algorithm, which is 5.82 W/h. The second row of the table shows the energy for the system with the energy consumption algorithm, which is 5.13 W/h. The table shows that the system with an energy consumption algorithm has a lower energy increment than the system without an energy consumption algorithm. This means that intelligently switching between proactive and reactive states of the nodes represents energy savings in the monitoring system.

**Table 7.** Energy consumption comparison.

| System Outline | Energy (W/h) |
|---|---|
| Without energy consumption algorithm | 5.82 |
| With energy consumption algorithm | 5.13 |

Resilience WSNs refers to the network's ability to maintain its functionality and performance even in the face of various challenges, disturbances, or failures. These challenges can include node failures, communication link disruptions, environmental changes, or malicious attacks. A resilient WSN can adapt to changes and recover quickly from disruptions, ensuring the continuous and reliable operation of the network. The concept of resilience in WSNs is crucial because these networks are often deployed in dynamic and harsh environments where failures or disruptions are common. For example, in environmental monitoring applications, WSNs may be deployed in remote and inaccessible locations where nodes may fail due to harsh weather conditions or energy depletion. The ability of a network to return to its stable state after facing a disruption is known as "time to recovery" or "time to convergence". It measures the duration it takes for the network to recover its normal functioning and reach a stable state after a failure or disturbance. A shorter time to recovery indicates a more resilient network that can quickly adapt to changes and restore its operations.

Figure 2 shows the distribution of the grid resilience with and without the energy algorithm. Thanks to the user's recommendations, the network should present fewer sudden changes in the logical topology. The logical topology is based on giving priority to nodes under proactive nature and less priority to nodes under reactive nature. We observe that under the energy algorithm, the network is 8% more resilient than when the algorithm was implemented. This situation may be because the network is adapting the nature of its nodes, and the recommendations given to the user are optimizing the person's habits. This is reflected in the fact that we have increasingly fewer alerts given by the system, and the person's sleep may tend to improve.

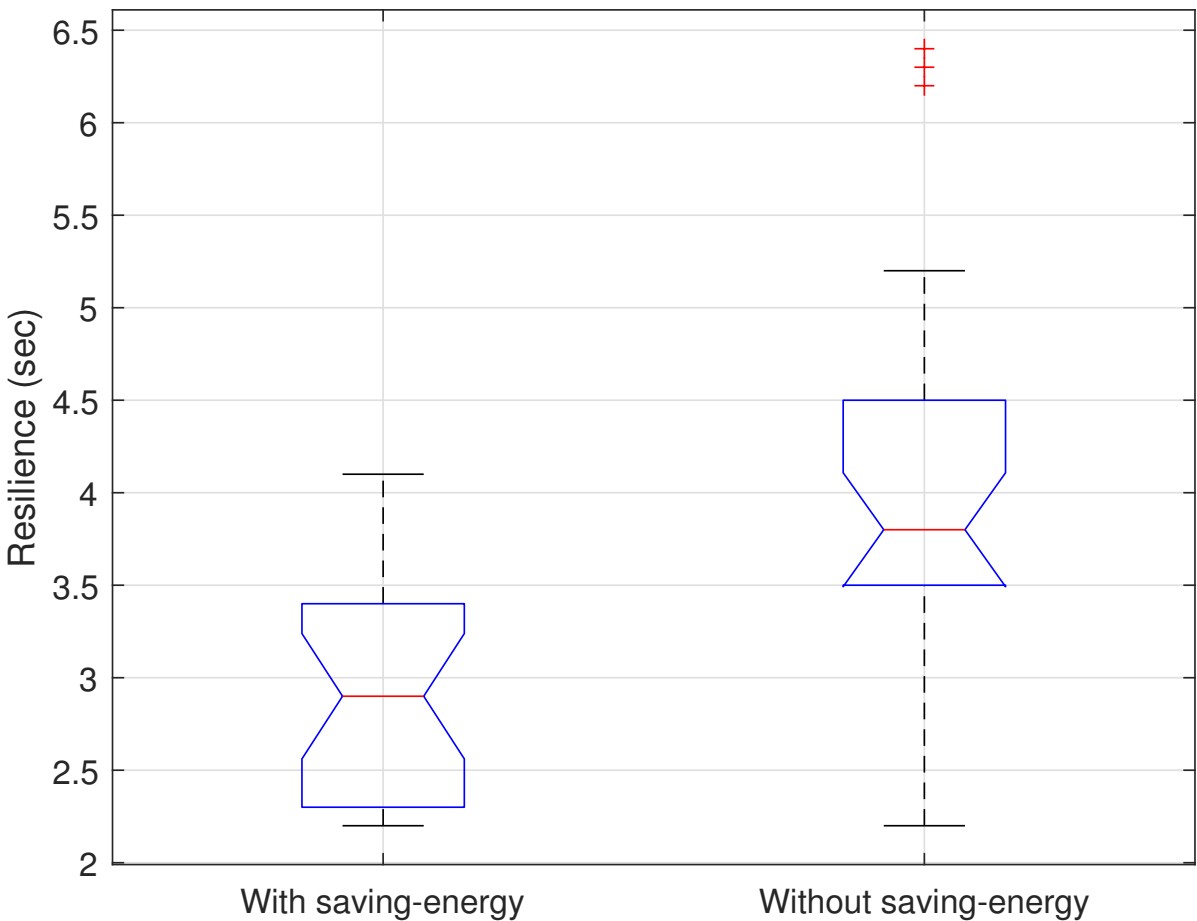

**Figure 2.** System resilience against changes in user habits.

Table 3 provides a useful comparison considering other related works devoted to develop ad hoc mathematical optimization models and algorithms for optimal energy management in WSN. This scheme presents a comparison of state-of-the-art energy-saving algorithms for remote monitoring of older adults at home. It lists seven algorithms, including Sleep-Scheduler, Energy-Aware Tree-Based Routing, Distributed Collaborative Routing, Multi-Objective Optimized Routing, Adaptive Clustering with QoS, Adaptive Compression, and QoI-aware. It includes a range of approaches and metrics, allowing interested readers to compare and evaluate the relative strengths and weaknesses of each algorithm.

Figure 3 shows the average number of alerts that are presented per day during the two weeks of experimentation with the algorithm in the monitoring system. We noticed that thanks to the follow-up of recommendations by the user, the system presents fewer alerts each day. This fact shows that the system has an acceptable and receptive degree of adaptation toward the user. In addition, if the user respects these recommendations, the system can guarantee a degree of improvement in the conditions to improve a person's sleep at home. The sensors send the alerts via email or SMS to the person to show them if a sensor presents more significant variation than usual. This is so that the person knows that behavior is not within the established thresholds in that area of the house. For example, a more significant variation in air quality, higher abnormal noise levels, or greater use of particular doors at night means that the person is not resting properly, etc. The purpose of

evidencing these alerts from the sensor system is that they should decrease as the system provides the person with recommendations daily.

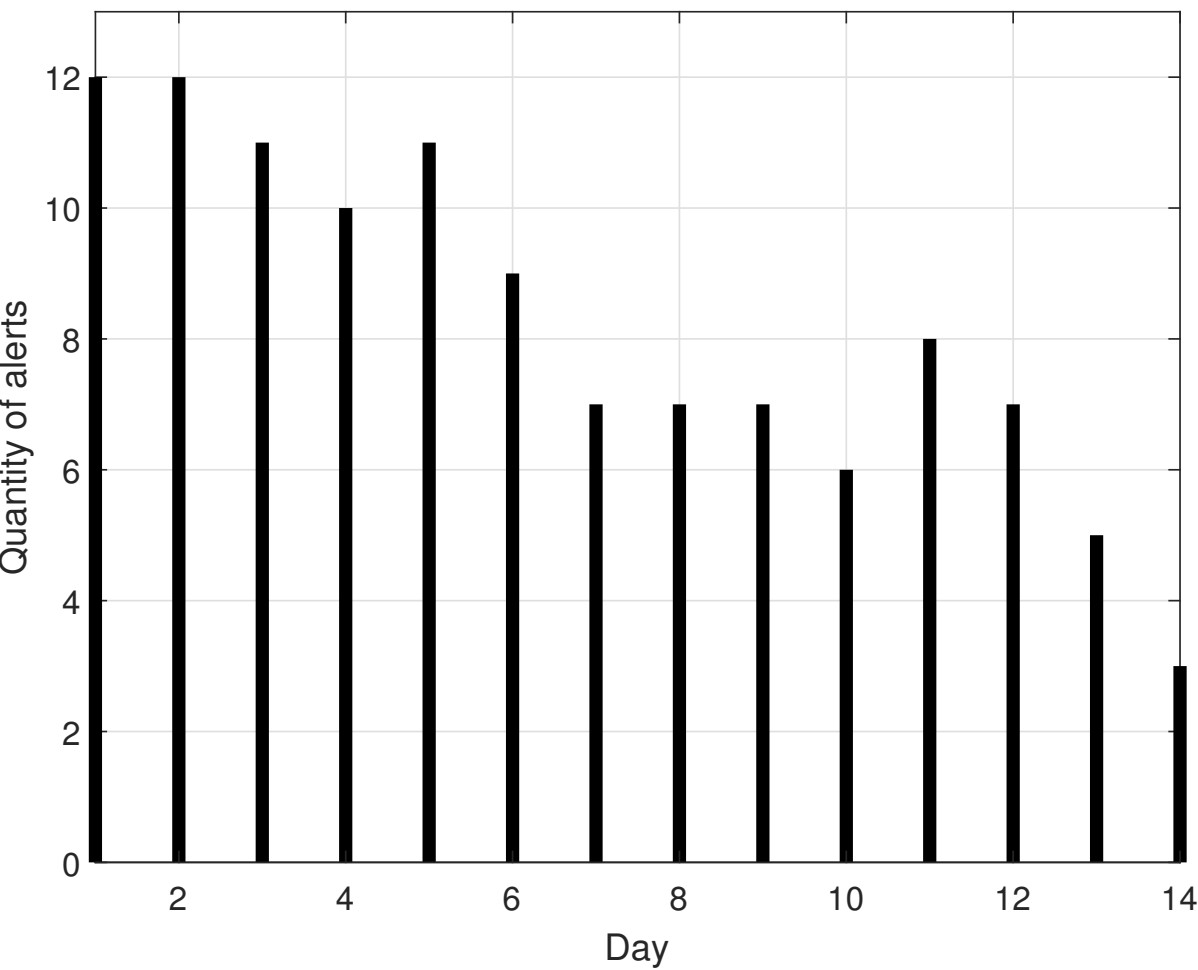

**Figure 3.** Number of alerts as the days go by with the proposed system.

### 4.2. Results of the Sensor Network over the Sleep Metrics

This section analyses the results of the sensor network over the sleep metrics of 30 older adults. The test includes the following metrics: deep sleep (DS), which describes the percent of deep sleep in the total sleep; REM sleep (REMS), which describes the percent of REM Sleep in the total sleep; heart rate (HR), which accounts for the number of beats per minute; breathing frequency (BF), which accounts for the number of breaths per minute; temperature (T), which describes the body temperature; oxygen saturation (OS), which measures the amount of oxygen in the blood; Heart rate variability (HRV), is an index of body relaxation, where a higher HRV indicates a relaxed and less stressed state than a lower HRV; total sleep time (TST) in hours; and snoring (S), which accounts for the waking hours between sleep time. Metrics were measured for four weeks, the first two weeks without the use of the sensor network and the last two weeks with its use. Table 8 shows the statistical description of the metrics with and without the use of the sensor network.

**Table 8.** Statistical description of the metrics gathering without and with the sensor network.

| | Without Sensor Network | | |
|---|---|---|---|
| **Metric** | **Mean** | **Median** | **St. Dev.** |
| DS | 13.11 | 12.5 | 4.04 |
| HR | 81.14 | 82.5 | 14.459 |
| BF | 16.78 | 17 | 2.766 |
| T | 37.319 | 37.3 | 0.2604 |
| REMS | 19.64 | 20 | 2.153 |
| OS | 93.94 | 94 | 3.132 |
| HRV | 9.92 | 10 | 2.662 |
| S | 8.47 | 8 | 3.81 |
| TST | 5.77 | 6 | 0.737 |
| | With Sensor Network | | |
| **Metric** | **Mean** | **Median** | **St. Dev.** |
| DS | 15.24 | 15 | 3.137 |
| HR | 75.82 | 75 | 10.166 |
| BF | 14.59 | 15 | 1.837 |
| T | 36.812 | 36.8 | 0.2653 |
| REMS | 22.53 | 22 | 1.66 |
| OS | 95.23 | 95 | 2.304 |
| HRV | 15.98 | 16.5 | 2.902 |
| S | 5.62 | 6 | 2.943 |
| TST | 7.2 | 7 | 0.816 |

4.2.1. Paired Hypothesis Tests Analysis

The first step was to determine if the metrics were significantly different with the use of the sensor network. To compare the results of each metric, it was necessary to know if the sample differences (with the sensor network—without the sensor network) came from a normal distribution for deciding the corresponding hypothesis test. For differences that came from a normal distribution, the appropriate test is the Paired *t*-test otherwise, we have to use the Wilcoxon test.

We plotted a Histogram and a Q-Q plot to verify whether the metrics are normally distributed. Figure 4 shows the histogram and Q-Q plots for the metrics differences (with the sensor network—without the sensor network).

It is not clear whether the metrics follow a normal distribution using the graphic method. In this case, we performed a statistical normality test to take more evidence. The most common normality tests are Shapiro–Wilk and Kolmogorov–Smirnov, Shapiro–Wilk is used when the sample size is equal to 50 or lower. The Kolmogorov–Smirnov test is used when the sample size is larger than 50. Because the sample size is larger than 50 we used the Kolmogorov–Smirnov test to prove if the metrics differences follow a normal distribution. The null hypothesis H0 for the Kolmogorov–Smirnov test is that the metrics follow a normal distribution. If the *p*-value of the Kolmogorov–Smirnov test is lower than 0.05 the H0 is rejected, otherwise, the data are normally distributed. Table 9 shows the *p*-value of the Kolmogorov–Smirnov test for the nine studied metrics' differences and whether those follow a normal distribution.

**Table 9.** Results of Kolmogorov–Smirnov test metrics samples differences (With the sensor network—without the sensor network).

| **Metric** | ***p*-Value** | **Normal Distribution** | **Appropriate Test** |
|---|---|---|---|
| DS | 0.001 | No | Wilcoxon |
| HR | 0.135 | Yes | *t*-test |
| BF | 0.077 | Yes | *t*-test |
| T | 0.021 | No | Wilcoxon |
| REMS | 0.001 | No | Wilcoxon |
| OS | 0.012 | No | Wilcoxon |
| HRV | 0.108 | Yes | *t*-test |
| S | 0.000 | No | Wilcoxon |
| TST | 0.000 | No | Wilcoxon |

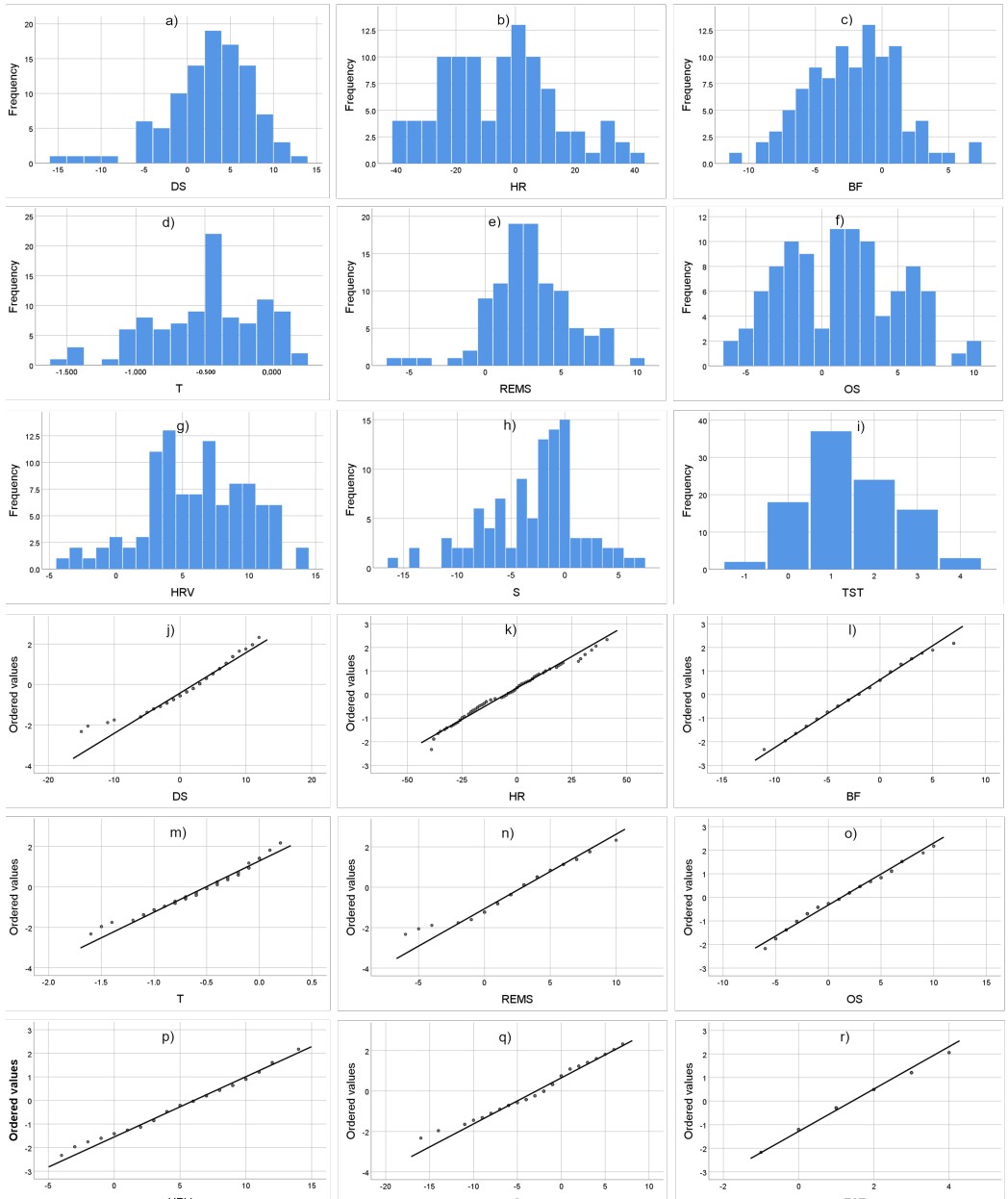

**Figure 4.** Figures (**a**–**i**) show the Histograms of the deep sleep (DS), heart rate (HR), breathing frequency (BF), temperature (T), REM sleep (REMS), oxygen saturation (OS), Heart rate variability (HRV), snoring (S), and, total sleep time (TST) metrics differences (with the sensor network—without the sensor network), respectively. And figures (**j**–**r**) show the Q-Q plot differences of the same nine sleep metrics.

The sample difference in heart rate, breathing frequency, and heart rate variability are normally distributed. On the other hand, the differences in deep sleep, REM sleep, temperature, oxygen saturation, total sleep time, and snoring are not normally distributed. Afterward, we performed the appropriate paired test for each variable. For heart rate, breathing frequency, and heart rate variability, we performed the paired *t*-test. For the rest of the variables, we performed the paired Wilcoxon test. For both tests, the H0 hypothesis is that the variables are not statistically different. In addition, for both tests, if the *p*-value is lower than 0.05 the data are significantly different with a 95% of the confidence interval, and the H0 is rejected. Table 8 shows the results of comparing the metrics with the use of the sensor network versus those without the use of the sensor network and if these are significantly different or not. In addition, Table 10 shows the result direction, SN > NSN

means that the correspondent metric is higher with the use of the sensor network, and SN < MNS means that the metric is lower with the use of the network.

**Table 10.** Results of paired tests between metrics with and without the use of the sensor network.

| Metric | *p*-Value | Test | Statistically Different | Difference Direction |
|---|---|---|---|---|
| DS | 0.000 | Wilcoxon | Yes | $SN > NSN$ |
| HR | 0.006 | *t*-test | Yes | $SN < NSN$ |
| BF | 0.000 | *t*-test | Yes | $SN < NSN$ |
| T | 0.000 | Wilcoxon | Yes | $SN < NSN$ |
| REMS | 0.000 | Wilcoxon | Yes | $SN > NSN$ |
| OS | 0.004 | Wilcoxon | Yes | $SN > NSN$ |
| HRV | 0.000 | *t*-test | Yes | $SN > NSN$ |
| S | 0.000 | Wilcoxon | Yes | $SN < NSN$ |
| TST | 0.000 | Wilcoxon | Yes | $SN > NSN$ |

It is possible to show that the nine metrics are different in a statistically significant way with the use of the sensor network.

### 4.2.2. Correlational Analysis

We then performed a correlation analysis between the metrics for both scenarios, with and without the use of the sensor network. At this point, to perform the appropriate correlation coefficient we examined whether each separate metric follows a normal distribution or not. If a couple of metrics from the same scenario (with the sensor network or without the sensor network) follow a normal distribution we can use a parametric correlation coefficient. Otherwise, we have to use a non-parametric correlation coefficient. As in the paired hypothesis, test analysis with the graphical methods was difficult to determine if the metrics are normally distributed, and we used a statistical normality test. Tables 11 and 12 show the results of the Kolmogorov–Smirnov test for the nine metrics for each scenario.

**Table 11.** Results of Kolmogorov–Smirnov test for metrics without the sensor network.

| Metric | *p*-Value | Normal Distribution |
|---|---|---|
| DS | 0.000 | No |
| HR | 0.021 | No |
| BF | 0.037 | No |
| T | 0.000 | No |
| REMS | 0.010 | No |
| OS | 0.000 | No |
| HRV | 0.008 | No |
| S | 0.003 | No |
| TST | 0.000 | No |

**Table 12.** Results of Kolmogorov–Smirnov test for metrics with the sensor network.

| Metric | *p*-Value | Normal Distribution |
|---|---|---|
| DS | 0.014 | No |
| HR | 0.002 | No |
| BF | 0.000 | No |
| T | 0.000 | No |
| REMS | 0.010 | No |
| OS | 0.000 | No |
| HRV | 0.000 | No |
| S | 0.004 | No |
| TST | 0.000 | No |

As the nine metrics with and without the use of the sensor network do not follow the normal distribution, we used the Spearman correlational coefficient to examine the correlations between each pair of metrics in both scenarios (with and without the sensor

network). The following statistically significant correlations in the metrics without the use of the sensor network scenario were found:

1.  Temperature and heart rate present negative correlations of $-0.331$ with a *p*-value of 0.001.
2.  Temperature and REM sleep present a positive correlation of 0.221 with a *p*-value of 0.027.
3.  Breathing frequency and heart rate variability present a positive correlation of 0.211 with a *p*-value of 0.035.

On the other hand, with the use of the sensor network the following statistically significant correlations were found:

1.  Temperature and REM sleep present a negative correlation of $-0.314$ with a *p*-value of 0.001.
2.  Total sleep time and snoring present a negative correlation of $-0.282$ y with a *p*-valor of 0.004.

### 4.2.3. Cluster Analysis

We performed a cluster analysis with the metrics to examine how the samples were divided into clusters before we used the sensor network and how they stayed in the same cluster or change to another cluster after the sensor network is used. With this information, it is possible to find the impact tendencies of the sensor network on the samples.

As for gathering samples without the sensor network as well as those gathering with the sensor network, the best number of clusters is 3 with 0.556 and 0.562 silhouette scores, respectively. With the use of the sensor network, the three clusters present an improvement in comparison with the clusters formed without the use of the sensor network. We classified and labeled the samples that changed to a better performance cluster, the samples that stayed in the same cluster, and the samples that changed to a worse performance cluster using the sensor network. Taking into account that generally the nine metrics improve with the sensor network we used the label's high impact for the samples that changed to a better cluster, the middle impact for samples that stay in the same cluster, and low impact for samples that changed to a worse cluster. There are 31 samples labeled as high impact, 42 samples as middle impact, and 27 as low impact. Labeling samples based on changes in metric performance is essential to perform regression analysis and discover predictive patterns in using the sensor network. Additionally, it allows analysis of differences between metrics before and after the use of the sensor network, for each group of samples isolated by label.

### 4.2.4. Sub-Grouping Analysis

We analyzed the metrics differences without and with the use of the sensor network for samples labeled as high impact, middle impact, and low impact in a separate way. As we did in the analysis of paired tests section, we corroborate whether the metric differences (with the sensor network—without the sensor network) of each class (high, middle, and low impact) follow a normal distribution to perform the appropriate paired test. The subgroups present a difficult normality distribution interpretation using the graphical methods, which means we had to use a statistical test to determine it. As the number of samples for the classes is lower than 50 we used the Shapiro–Wilk test. Tables 13–15 show the results of the Shapiro–Wilk test and the corresponding pair test to use. Tables 16–18 show the results of the paired tests for each impact class and the directions of the results.

We found that samples labeled as high impact have different statistical significance in the nine metrics. For the samples labeled as middle impact, only the oxygen saturation (OS) metric does not have a significant difference with the use of the sensor network. Moreover, in the low impact group, the metrics temperature, REM sleep, total sleep time, heart rate variability, and snoring show a significant difference, but deep sleep, heart rate, breathing frequency, and oxygen saturation do not present a significant difference with the use of the sensor network. It is possible to see that metrics temperature, REM sleep, total sleep time,

heart rate variability, and snoring exhibit a significant difference with the use of the sensor network in any labeled group.

**Table 13.** Results of Shapiro–Wilk test metrics samples differences (with the sensor network—without the sensor network) for the class high impact.

| Metric | *p*-Value | Normal Distribution | Appropriate Test |
|---|---|---|---|
| DS | 0.912 | Yes | *t*-test |
| HR | 0.391 | Yes | *t*-test |
| BF | 0.374 | Yes | *t*-test |
| T | 0.852 | Yes | *t*-test |
| REMS | 0.121 | Yes | *t*-test |
| OS | 0.234 | Yes | *t*-test |
| HRV | 0.260 | Yes | *t*-test |
| S | 0.787 | Yes | *t*-test |
| TST | 0.004 | No | Wilcoxon |

**Table 14.** Results of Shapiro–Wilk test metrics samples differences (with the sensor network—without the sensor network) for the class middle impact.

| Metric | *p*-Value | Normal Distribution | Appropriate Test |
|---|---|---|---|
| DS | 0.004 | No | Wilcoxon |
| HR | 0.644 | Yes | *t*-test |
| BF | 0.495 | Yes | *t*-test |
| T | 0.125 | Yes | *t*-test |
| REMS | 0.048 | No | Wilcoxon |
| OS | 0.254 | Yes | *t*-test |
| HRV | 0.230 | Yes | *t*-test |
| S | 0.027 | No | Wilcoxon |
| TST | 0.016 | No | Wilcoxon |

**Table 15.** Results of Shapiro-will test metrics samples differences (with the sensor network—without the sensor network) for the class low impact.

| Metric | *p*-Value | Normal Distribution | Appropriate Test |
|---|---|---|---|
| DS | 0.017 | No | Wilcoxon |
| HR | 0.316 | Yes | *t*-test |
| BF | 0.903 | Yes | *t*-test |
| T | 0.186 | Yes | *t*-test |
| REMS | 0.005 | No | Wilcoxon |
| OS | 0.154 | Yes | *t*-test |
| HRV | 0.031 | No | Wilcoxon |
| S | 0.550 | Yes | *t*-test |
| TST | 0.003 | No | Wilcoxon |

**Table 16.** Results of paired tests between metrics with and without the use of the sensor network for the class high impact.

| Metric | *p*-Value | Test | Statistically Different | Difference Direction |
|---|---|---|---|---|
| DS | 0.001 | *t*-test | Yes | SN > NSN |
| HR | 0.000 | *t*-test | Yes | SN < NSN |
| BF | 0.000 | *t*-test | Yes | SN < NSN |
| T | 0.000 | *t*-test | Yes | SN < NSN |
| REMs | 0.000 | *t*-test | Yes | SN > NSN |
| OS | 0.000 | *t*-test | Yes | SN > NSN |
| HRV | 0.000 | *t*-test | Yes | SN > NSN |
| S | 0.000 | *t*-test | Yes | SN < NSN |
| TST | 0.000 | Wilcoxon | Yes | SN > NSN |

**Table 17.** Results of paired tests between metrics with and without the use of the sensor network for the class middle impact.

| Metric | *p*-Value | Test | Statistically Different | Difference Direction |
|:---:|:---:|:---:|:---:|:---:|
| DS | 0.001 | Wilcoxon | Yes | $SN > NSN$ |
| HR | 0.003 | *t*-test | Yes | $SN < NSN$ |
| BF | 0.000 | *t*-test | Yes | $SN < NSN$ |
| T | 0.000 | *t*-test | Yes | $SN < NSN$ |
| REMS | 0.000 | Wilcoxon | Yes | $SN > NSN$ |
| OS | 0.623 | *t*-test | No | $SN > NSN$ |
| HRV | 0.000 | *t*-test | Yes | $SN > NSN$ |
| S | 0.022 | Wilcoxon | Yes | $SN < NSN$ |
| TST | 0.000 | Wilcoxon | Yes | $SN > NSN$ |

**Table 18.** Results of paired tests between metrics with and without the use of the sensor network for the class low impact.

| Metric | *p*-Value | Test | Statistically Different | Difference Direction |
|:---:|:---:|:---:|:---:|:---:|
| DS | 0.264 | Wilcoxon | No | $SN > NSN$ |
| HR | 0.130 | *t*-test | No | $SN > NSN$ |
| BF | 0.494 | *t*-test | No | $SN < NSN$ |
| T | 0.000 | *t*-test | Yes | $SN < NSN$ |
| REMS | 0.016 | Wilcoxon | Yes | $SN > NSN$ |
| OS | 0.901 | *t*-test | no | $SN > NSN$ |
| HRV | 0.000 | Wilcoxon | Yes | $SN > NSN$ |
| S | 0.026 | *t*-test | Yes | $SN < NSN$ |
| TST | 0.000 | Wilcoxon | Yes | $SN > NSN$ |

### 4.2.5. Regression Analysis

At this point, we know how the metrics are impacted by the sensor network on a qualitative scale (low, middle, and high impact). With this information, we could perform a regression analysis and identify predictive tendencies using the nine metrics gathered without the use of the sensor network as independent variables and the tag (high impact, middle impact, and low impact) as the target variable. In this way, we performed a multinomial logistic regression to discover how effective the sensor network can be for improving the sleep metrics once we know the nine metrics without the use of the sensor network. Because the tag is a nominal variable, it is good practice to take the first class as a reference to estimate the parameters, in this analysis we took the low impact as the reference class. Taking the low impact class as a reference class for the estimation of parameters clarifies the visualization of impact in the use of the sensor network over the nine metrics because in this way we can compare the possibilities to have a middle impact or high impact versus a low impact.

Table 19 shows the estimation parameters with the reference class low impact of the multinomial logistic regression made with a steep back Wald test methodology.

Using a steep back Wald test methodology in the multinomial logistic regression, it was possible to appreciate that the heart rate variability and deep sleep metrics are not in the estimation of the parameters of Table 19. In the estimation of the multinomial logistic regression, if the p-value of a metric is higher than 0.05 in the Wald test, the variable modifies less than 5 percent of the variance of the class to predict. The steep-back Wald test methodology performs the model only with the variables that predict more than 5% of the predicted class variance, which means that heart rate variability and deep sleep have *p*-values above 0.05 for the two predicted classes.

**Table 19.** Significant estimation parameters result from the multinomial regression with the reference class low impact.

| Predicted Class | Metric | Coefficient | *p*-Value | Odd Radios |
|---|---|---|---|---|
| High impact | Intersection | −133.579 | 0.040 | |
| | HR | 0.174 | <0.001 | 1.190 |
| | BF | 0.607 | <0.001 | 1.835 |
| | T | 4.877 | 0.007 | 131.209 |
| | REMS | −0.380 | 0.073 | 0.684 |
| | OS | −0.652 | <0.001 | 0.521 |
| | TST | −1.180 | 0.028 | 0.307 |
| | S | 0.357 | 0.005 | 1.429 |
| Middle impact | Intersection | −106.781 | 0.062 | |
| | HR | 0.097 | 0.003 | 1.102 |
| | BF | 0.330 | 0.007 | 1.391 |
| | T | 3.249 | 0.039 | 25.762 |
| | REMS | −0.475 | 0.008 | 0.622 |
| | OS | −0.162 | 0.186 | 0.850 |
| | TST | −0.430 | 0.279 | 0.650 |
| | S | 0.053 | 0.579 | 1.054 |

For the class high impact, the metrics heart rate, breathing frequency, temperature, oxygen saturation, total sleep time, and snoring are significant predictors, and the metric REM sleep is a non-significant predictor. Also, for the class middle impact the metrics heart rate, breathing frequency, temperature, and REM sleep are significant predictors, and the metrics oxygen saturation, total sleep time, and snoring are non-significant predictors.

The multinomial logistic regression model has a coefficient of Cox and Snell of 0.536 and a coefficient of Nagelkerke of 0.65 indicating that the independent variables explain approximately 60 percent of the dependent variable variance. On the other hand, the model has a McFadden coefficient of 0.355 indicating good fitness.

## 5. Discussion

Remote monitoring systems based on sensor networks in smart homes for the elderly have the potential to revolutionize the way care is provided to this population. By using a network of wireless sensors to gather data on the individual's activity and environment, these systems can provide valuable insights into the individual's health and well-being. The use of wearable sensors to monitor vital signs such as heart rate and oxygen saturation, in particular, can provide early warning signs of potential health issues, allowing for prompt intervention and medical attention. This can be especially important for older adults who may be at risk of falls or other accidents, as well as those with chronic conditions such as heart disease or diabetes.

Furthermore, the use of machine learning and artificial intelligence techniques to analyze the data collected by the sensors can also provide additional benefits. These technologies can be used to detect patterns and anomalies in the individual's behavior or vital signs, which can provide early warning signs of potential issues. This can include identifying changes in sleep patterns that may indicate the onset of a sleep disorder, or detecting a decline in mobility that may increase fall risks. Additionally, these systems can also be used to monitor medication adherence, which can be especially important for older adults who may have difficulty remembering to take their medication.

In addition, the data collected by the systems can also be used to provide more personalized care, based on the individual's specific needs and preferences. By monitoring the individual's activities and movements, for instance, the systems can identify patterns of behavior that can be used to adjust the individual's care plan. This can include adjusting the lighting or temperature in the home, or providing reminders to take medication or perform other important tasks.

Overall, remote monitoring systems with sensor networks in smart homes for the elderly can provide a wide range of benefits, including improved health outcomes, increased safety and security, and more personalized care. By tracking vital signs and activity,

these systems can provide early warning signs of potential issues, allowing for prompt intervention and medical attention. Additionally, the use of machine learning and artificial intelligence techniques can provide additional benefits, such as identifying patterns and anomalies in the individual's behavior or vital signs, and providing more personalized care based on the individual's specific needs and preferences.

A remote sensor network monitoring system can have a significant impact on alarm feedback and recommendations for users to improve their sleeping habits. The system allows for continuous monitoring of sleep patterns and provides real-time data on sleep efficiency, latency, wake after sleep onset, and total sleep time. These data can be used to provide personalized alarm feedback and recommendations to the user to improve their sleep. For example, if the system detects that the user has a high sleep latency or wake after sleep onset, it can provide recommendations to the user to improve their sleep hygiene, such as avoiding caffeine or electronic devices before bedtime.

Additionally, the system can also provide recommendations for the user to improve their sleep environment. For example, if the system detects that the user's room temperature is too high or too low, it can provide recommendations to adjust the temperature to optimize sleep. It can also detect noise, light and humidity level, as well as other environmental factors that can affect sleep quality, providing feedback to the user to minimize or eliminate these factors. Furthermore, it can also detect the use of sleeping aids or any other drug use, and make recommendations if needed. Overall, the remote sensor network monitoring system provides users with valuable insights into their sleep patterns and ways to improve their sleeping habits, which can lead to better overall sleep health and quality of life.

This study is based on an original proposal for a sensor system that adapts to the person's habits using a low energy-consumption algorithm. The sensor system sends recommendations to users daily. These recommendations are tailored to a person's ideal sleeping conditions. Recommendations are based on the person's use when moving through different areas of their house. The sleep improvement information is verified daily through a smartwatch to identify whether the system is having a positive, negative or neutral impact on the habits of falling asleep. However, if the person truly has a sleep disorder or is regularly interrupted, using this sensor system will not be a reliable indicator of sleep health. Of course, the best option will be a sleep study that a physician can order to obtain a reliable and accurate reading.

With the results of the paired hypothesis test analysis, the correlation analysis, and the cluster analysis that includes a sub-groping and a regression analysis over the sleep studied metrics we can discuss the following findings.

In sleep, the body enters a relaxed state, and the health metrics change. For example, during the sleep period the heart rate, breathing frequency, and temperature tend to decrease [51,52]. Also, for an adult, the recommended total sleep time ranges from 7 to 8 h [53] with approximately 20% of deep sleep and 25% REM sleep for quality rest. In the paired hypothesis test analysis, it was possible to see that with the use of the sensor network, the nine metrics were statistically different, total sleep time, deep sleep, REMS sleep, heart rate variability, and oxygen saturation increased, while, heart rate, breathing frequency, temperature, and snoring decreased. These results suggest that the nine metrics improved in a general way with the use of the sensor network.

Although the correlations found in the correlation analysis are not strong, some of them show some tendencies. Once knowing that the total sleep time increased and the snoring decreased, the negative correlation between the total sleep time and snoring with the use of the sensor network can mean that elderly people slept more and with less interruptions with the use of the sensor network. Temperature and REM sleep change the direction of the correlation with the use of the sensor network, they pass from a positive correlation to a negative correlation. Once knowing that the temperature decreased and the REM sleep increased the negative correlation can mean that the REM sleep increases when the temperature decreases. According to [52], body temperature decreases in the REM sleep states, nevertheless, REM sleep occurs over a much longer duration than deep sleep,

which can be why REM sleep has a correlation with temperature in this study and deep sleep does not. Finally, the weak correlations between heart rate with temperature and breathing frequency with heart rate variability that represented a contradictory behavior as to what the metrics would take in a quality rest disappeared with the use of the sensor network. This can mean that the sensor network stabilizes the behavior of the heart rate, temperature, breathing frequency, and heart rate variability.

With the estimation of parameters for the multinomial logistic regression made in the regression analysis, we could find the metrics with a predictive trend in the impact level that the use of the sensor network can have once known the sleep metrics without using this. Taking the low impact labeled class as a reference to calculate the odds ratio between the possibility to obtain a positive significant difference in the nine metrics (high impact) versus the possibility to have a positive significant difference in five metrics (temperature, REM sleep, total sleep time, and heart rate variability for the low impact class) we found that:

1. For each extra heartbeat per minute and keeping the intersection and the other variables constant, the possibility of high impact increases 1.90 times.
2. For each extra respiration per minute and keeping the intersection and the other variables constant, the possibility of high impact increases 1.835 times.
3. For each additional degree Celsius in body temperature and keeping the intersection and the other variables constant, the possibility of high impact increases 131.209 times. The increase is large because a Celsius grade is a big difference for a temperature body change.
4. For each additional unit of oxygen saturation and keeping the intersection and the other variables constant, the possibility of high impact decreases 1.919 times.
5. For each extra hour of total sleep time and keeping the intersection and the other variables constant, the possibility of high impact decreases 3.257 times.
6. For each extra hour of snoring and keeping the intersection and the other variables constant, the possibility of high impact increases 1.429 times.

In the odds ratios calculation between the possibility to obtain a positive significant difference in eight of the nine metrics (except oxygen saturation for the middle impact class) versus the possibility to have a positive significant difference in five metrics (temperature, REMS sleep, total sleep time, and heart rate variability for the low impact class) we found that:

1. For each additional heartbeat per minute and keeping other variables constant, the possibility of middle impact increases 1.102 times.
2. For each additional respiration per minute and keeping other variables constant, the possibility of high impact increases 1.291 times.
3. For each additional degree Celsius in body temperature and keeping the other variables constant, the possibility of high impact increases 25.762 times.
4. For each additional hour of REMS sleep time and keeping the intersection and the other variables constant, the possibility of high impact decreases 1.607 times.

In the estimation of parameters for both, high- and middle-impact labeled classes, there is a common tendency in three metrics. The higher the heart rate, breathing frequency, and temperature without the use of the sensor network, the possibility of better results with the use of the sensor network increase.

## 6. Conclusions

The remote sensor network monitoring system is a valuable tool for improving sleep habits and health. The system provides real-time data on sleep patterns and offers personalized feedback and recommendations based on these data. The system can detect and analyze a wide range of factors that impact sleep quality, including sleep efficiency, latency, wake after sleep onset, temperature, noise, light, humidity levels, and the use of sleeping aids or medications. These recommendations are tailored to the individual's habits and

sleep environment and are verified daily through a smartwatch. However, it is important to note that if a person has a sleep disorder or regularly experiences interruptions, the system may not provide a reliable indicator of sleeping health and a physician-prescribed sleep study may be necessary.

In this work, we evaluated the impact of a remote sensor network monitoring system on sleep patterns and habits. The results showed that the system had a statistically significant impact on nine metrics related to sleep, including total sleep time, deep sleep, REM sleep, heart rate variability, oxygen saturation, heart rate, breathing frequency, temperature, and snoring. The study also found that the sensor network increased the total sleep time in elderly adults making the sleep less interrupted while stabilizing temperature, REM sleep, heart rate, breathing frequency, and heart rate variability.

However, it is important to note that while the results of this study indicate that the sensor network can have a positive impact on sleep, it may not be a reliable indicator for individuals with sleep disorders or those who are regularly interrupted. A sleep study ordered by a physician remains the best option for an accurate and reliable reading. Nevertheless, the sensor network is an innovative solution that provides users with valuable insights into their sleep patterns and ways to improve their sleeping habits, which can lead to better overall sleep health and quality of life.

The cluster analysis performed in the study revealed that the use of the sensor network has a positive impact on various sleep metrics such as temperature, REM sleep, total sleep time, and heart rate variability for the samples that were grouped into high impact, middle impact, and low impact subgroups. Further regression analysis provided insight into the predictive trends of the sensor network's impact level on sleep metrics. The results showed that factors such as heart rate, breathing frequency, body temperature, REM sleep, oxygen saturation, total sleep time, and snoring have a significant impact on the level of improvement with the use of the sensor network. Additionally, the study found that higher heart rate, breathing frequency, and body temperature without the use of the sensor network increase the possibility of better results when the sensor network is used.

**Supplementary Materials:** The following supporting information can be downloaded at: https://www.mdpi.com/article/10.3390/fi15090287/s1.

**Author Contributions:** C.D.-V.-S. developed the analysis, supervised the research methodology and the approach of this work, prepared the scenario and analyzed the results. R.A.B. reviewed, interpreted and drafted the comparison results and tables. L.J.V. was involved on the formal analysis and the manuscript. R.V. made the figures and their comparison and performed the formal analysis. J.A.N.-F. directed some formal concepts and review the manuscript. All authors have read and agreed to the published version of the manuscript.

**Funding:** This research received no external funding.

**Institutional Review Board Statement:** Integrity Code of the Universidad Panamericana, validated by the Social Affairs Committee and approved by the Governing Council through resolution CR 98-22, on 15 November 2022.

**Informed Consent Statement:** Written informed consent was obtained from the patient(s) to publish this paper.

**Data Availability Statement:** Data availability requests can be sent to the Journal through the Supplementary Materials.

**Conflicts of Interest:** The authors declare no conflicts of interest.

## Appendix A. Equations

*Appendix A.1. t-Test*

$$t = \frac{\bar{d}}{s_d / \sqrt{n}} \tag{A1}$$

where:

    $t$: is the value of the Student's t statistic.

    $\bar{d}$: denotes the average of the differences observed between the related pairs of measurements.

    $s_d$: stands for the standard deviation of the differences between the related pairs of observations.

    $n$: indicates the sample size, which refers to the number of related pairs of observations.

*Appendix A.2. Wilcoxon Test*

$$W = \sum_{i=1}^{n} r_i \tag{A2}$$

where: $W$: is the Wilcoxon test statistic.

    $n$: indicates the sample size (number of related pairs of observations).

    $r_i$: is the rank of the $i$-th difference between the related observations.

*Appendix A.3. Shapiro–Wilk Test*

$$W = \frac{\left(\sum_{i=1}^{n} a_i(x_i)\right)^2}{\sum_{i=1}^{n}(x_i - \bar{x})^2} \tag{A3}$$

where:

    $W$: is the Shapiro–Wilk test statistic.

    $(x_i)$: represent the ordered values of the data set, used to calculate the test statistic $W$.

    $\bar{x}$: denotes the mean of the data, used to calculate the coefficients $a_i$ required for the calculation of $W$.

    $a_i$: is the coefficients calculated from the mean, variance, and covariance of the data.

*Appendix A.4. Kolmogorov–Smirnov Test*

$$D_n = \max |F_n(X_i) - \Phi(X_i)| \tag{A4}$$

where:

    $D_n$: is the test statistic for the Kolmogorov–Smirnov test.

    $F_n(X_i)$: is the empirical distribution function of the sample.

    $\Phi(X_i)$: is the cumulative distribution function for a standard normal distribution (with mean 0 and standard deviation 1).

    max: refers to the maximum absolute difference between the two distribution functions.

*Appendix A.5. Pearson Correlation Coefficient*

$$r_{xy} = \frac{\sum_{i=1}^{n}(x_i - \bar{x})(y_i - \bar{y})}{\sqrt{\sum_{i=1}^{n}(x_i - \bar{x})^2}\sqrt{\sum_{i=1}^{n}(y_i - \bar{y})^2}} \tag{A5}$$

where

    $r_{xy}$: is the Pearson correlation coefficient between two variables $x$ and $y$.

    $\bar{x}$ and $\bar{y}$: are the sample means of $x$ and $y$, respectively.

    $n$: is the number of observations in the sample.

*Appendix A.6. Spearman Correlation Coefficient*

$$\rho = 1 - \frac{6\sum_{i=1}^{n} d_i^2}{n(n^2 - 1)} \tag{A6}$$

where
    $\rho$: is the Spearman correlation coefficient between two variables.
    $d_i$: is the difference between the ranks of each pair of corresponding observations.
    $n$: is the number of observations in the sample.

*Appendix A.7. k-Means*

    The k-means algorithm seeks to partition n observations into K (K < n) set of G groups $G = \{G_1, G_2, G_3...G_K\}$ minimizing the within-cluster sum of squares.

$$arg\ min\ G \sum_{i=1}^{K} \sum_{x \in G_i} (x - \mu_i)^2 \tag{A7}$$

where:
    *arg min G*: is the minimum argument in the set of clusters G.
    $\mu_i$: is the mean of the values in the cluster $G_i$.

*Appendix A.8. Silhouette Value*

$$s_i = \frac{b_i - a_i}{\max\{a_i, b_i\}} \tag{A8}$$

where:
    $s_i$: is the silhouette value for the i-th object.
    $a_i$: is the average distance between the i-th object and all other objects in the same cluster.
    $b_i$: is the average distance between the i-th object and all objects in the nearest cluster (other than the cluster containing the i-th object).

*Appendix A.9. Multinomial Multivariable Logistic Regression with a Logit Parameter Estimation*

    In multinomial logistics regression, the probability of the set of C classes predicted is calculated, and the class possibilities are associated, which means the sum of all class possibilities is 1. Equation (A9) shows the mathematical description of the multinomial logistics regression function to calculate a class probability.

$$p_i = \frac{e^{z_i}}{\sum_{j=1}^{C} e^{z_j}} \tag{A9}$$

where:
    $p_i$: is the probability of class I.
    $z_i$: is denoted by the Equation (A10).

$$z_i = \alpha_{i1}x_1 + \alpha_{i2}x_2 + \alpha_{i3}x_3 ... + \alpha_{in}x_n + \beta_i \tag{A10}$$

where:
    $x_1, x_2, x_3... x_n$ represent the independent variables and n is the number of independent variables.
    Also, we took a reference class to estimate the parameter in odd ratios, Equation (A11) shows the transformed Logit function used to estimate parameters.

$$y = Logit(x) = \log \frac{p_{ij}}{p_{ir}} \tag{A11}$$

where:

$p_{ij}$: is the probability of sample i belonging to class j.
$p_{ir}$: is the probability that the same sample belongs to the reference class r.
$y$: is the multinomial predicted variable.

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
