# Peer review of "Non-Invasive Monitoring of Vital Signs for the Elderly Using Low-Cost Wireless Sensor Networks: Exploring the Impact on Sleep and Home Security"

_futureinternet, doi:10.3390/fi15090287_

Round 1

Reviewer 1 Report

Comments and Suggestions for Authors

1.       The main contributions of this work are not clearly highlighted. The paper should clearly give the environment, assumptions, requirements and objectives of the problem in hand, and points out major issues or difficulties when dealing with the problem.

2.         Vest amounts of existing studies have proposed the similar concept to monitor sleep behaviors using the WSNs. This paper should review the class of these papers and clearly give the differences and major improvements of the proposed mechanism. Thus, the literature gap filled by the present study is not clearly stated. Please expand on this statement.

3.          The discussion and evaluation part (Section IV) only considers several different parameter settings for the authors’ solutions and compares this work to the conventional schemes (i.e., with/without WSNs and with/without energy saving alg.). Therefore, it is unclear if and how this work advances the state of the art. Please expand on this statement.

4.          The connection among the sensors, security, the activities of the elder population is too weak. Please expand on this statement.   

5.          Spelling and grammar check is required.

Comments on the Quality of English Language

 Minor editing of English language is required.

Author Response

Dear

Editor

We are submitting the paper:

“Non-Invasive Monitoring of Vital Signs for the Elderly Using Low-Cost Wireless Sensor Networks: Exploring the Impact on Sleep and Home Security”

Authored by: Carolina Del-Valle-Soto *, Ramon A. Briseño, Leonardo J. Valdivia , Ramiro Velázquez , Juan Arturo Nolazco-Flores.

We would like to thank the reviewers and editors for their detailed analysis of the manuscript; the comments are very valuable to us. In the revised version of the paper, we have incorporated the all changes recommended by the reviewers.

Comments to all observations and suggestions including point-by-point responses are addressed in the following text.

Reviewer 1 comments

Comment 1: The main contributions of this work are not clearly highlighted. The paper should clearly give the environment, assumptions, requirements and objectives of the problem in hand, and points out major issues or difficulties when dealing with the problem.

Response: Many thanks to the Reviewer for his/her invaluable interest in the comments on this manuscript. We have added a paragraph in the Introduction that clarifies all the questions that the Reviewer properly raises.

The main contributions of this research work lie in developing and applying a wireless sensor network (WSN) for monitoring the vital signs of elderly individuals in their homes to improve their sleep quality. The environment is the same as the home of the elderly. The sensor network is installed in their home to be as strange or invasive as possible.

The study uses WSNs with low-cost sensors, including motion, pressure, temperature, humidity, noise, light, gyroscope, and air quality sensors, to gather real-time data from different house areas. The objective is to analyze the sleep metrics of older adults before and after implementing the sensor network to determine the impact of the system on improving rest. The proposed system aims to provide daily safety recommendations to the elderly and allow remote monitoring of vital signs, helping them to live independently and receive timely medical care. The research addresses the challenges of monitoring elderly individuals, especially in terms of their sleep patterns, as this age group tends to have reduced mobility and difficulty accessing health centers. The study demonstrates the potential of using WSNs and intelligent algorithms to optimize sensor utilization, improve sleep quality, and provide personalized recommendations for the elderly's well-being. The sensor system has a group of pre-written recommendations based on the possible causes of sensor parameter anomalies. These anomalies may be due to the person sleeping with much noise, getting up several times during the night, or the temperature in their home is too cold or too hot. They are only approximations of possible causes to improve the quality of life of people who live alone. This, being an approximation, is one of the system's limitations. The findings highlight the importance of continuous monitoring and non-invasive technology in enhancing the quality of life for older adults and preventing health complications. Overall, this work contributes to advancing remote monitoring systems for elderly individuals, particularly in the context of sleep monitoring and healthcare assistance.

Comment 2: Vest amounts of existing studies have proposed the similar concept to monitor sleep behaviors using the WSNs. This paper should review the class of these papers and clearly give the differences and major improvements of the proposed mechanism. Thus, the literature gap filled by the present study is not clearly stated. Please expand on this statement.

Response: Many thanks to the Reviewer. The Reviewer is right, and we have included a comparative table of representative works with advantages and limitations compared to our proposed system.

Table 2 provides a comprehensive comparison of various sleep monitoring mechanisms, including proposals from the literature and the state of the art. Each mechanism is evaluated based on its unique features and contributions. These existing mechanisms offer valuable insights into sleep monitoring, such as detecting sleep stages, assessing stress levels, monitoring sleep apnea, analyzing sleep patterns, observing sleep behavior, and tracking snoring. However, our proposed system, based on a non-invasive wireless sensor network (WSN) located in people's homes, introduces significant advancements in sleep monitoring technology. It operates without the need for wearables, making it more comfortable for users. The system monitors various parameters, such as luminosity, noise, and movement, providing a comprehensive view of the sleep environment and body parameters. By integrating multiple parameters, the proposed system can offer more accurate and personalized insights into sleep quality, generating personalized sleep recommendations. This non-wearable WSN in homes ensures a seamless and convenient monitoring experience, offering valuable improvements over existing mechanisms. Overall, the proposed system represents a promising and innovative approach to sleep monitoring, enhancing user comfort and providing actionable recommendations for better sleep quality.

Comment 3: The discussion and evaluation part (Section IV) only considers several different parameter settings for the authors’ solutions and compares this work to the conventional schemes (i.e., with/without WSNs and with/without energy saving alg.). Therefore, it is unclear if and how this work advances the state of the art. Please expand on this statement.

Response: Thanks to Reviewer. We have improved the explanation related to the energy saving of the algorithm, to make the paper more readable.

Based on Algorithm 1, the system presents an energy-saving concerning the regular operation of the wireless sensor network. For this result, we consider the system's use for one hour, and when we apply the algorithm, we observe that the use of the sensor batteries is reduced. This is only an added value to the system because its principal value is monitoring the conditions of the house to give recommendations based on sleep.

We have also added this paragraph to the state of the art to complement the progress aspects of the proposed work.

The work presented here advances the state of the art in proposing a sensor system for monitoring people at home and providing personalized recommendations for improving sleep quality based on anomalous levels of measurements. This system leverages sensor technology and data analysis to offer solutions to enhance sleep patterns. The development of a sensor system tailored for home monitoring is a significant step forward. This system likely consists of various types of sensors strategically placed within the living spaces to capture relevant data related to sleep patterns. These sensors could include ambient light sensors, temperature sensors, motion sensors, heart rate monitors, and even advanced devices like sleep-specific wearable technology. The work proposes a data analysis framework to process the data collected by the sensor system effectively. By analyzing and interpreting the data, the system can identify anomalous levels or patterns in a person's sleep-related measurements. Anomalies could include irregular sleep duration, abnormal heart rate variations during sleep, or disturbances in sleep cycles. The key innovation lies in the system's ability to provide personalized recommendations based on the anomalous measurements. These recommendations could range from changes in sleep habits, lifestyle adjustments, relaxation techniques, or even suggestions to consult a medical professional, depending on the severity of the anomaly detected. The energy-saving algorithm integrated into the sensor system is another crucial advancement. By utilizing the programming nature of the sensors (proactive and reactive), the system can efficiently manage power consumption and extend the overall lifespan of the sensors. This algorithm likely employs techniques like dynamic sampling, adaptive data transmission, or intelligent sleep/wake-up modes for the sensors, ensuring that they are only active when necessary.

Comment 4: The connection among the sensors, security, the activities of the elder population is too weak. Please expand on this statement. 

Response: The Reviewer's point is very coherent; therefore, we have added specific paragraphs in the Methodology, highlighting the contribution of these topics.

The connection between sensors, security, and the activities of the elderly is multifaceted:

Sensor System for Monitoring: The sensor system collects data from various sensors, including motion, pressure, temperature, humidity, noise, light, gyroscope, and air quality sensors. These sensors are designed to monitor the home environment and the elderly individual's vital signs while being unobtrusive and comfortable for the user.

Security and Privacy: The work emphasizes the need for non-invasiveness and respect for the privacy of the elderly. The data collected by the sensor network must be securely transmitted and stored to protect the individual's sensitive health information. Ensuring data security and privacy is crucial in healthcare applications, especially when dealing with sensitive data from vulnerable populations.

Activities of the Elderly: The sensor system aims to capture data related to the activities of the elderly in their homes. It tracks sleep patterns, movement, noise levels, and air quality to identify anomalies and provide personalized recommendations for improving sleep quality. By understanding the person's daily activities, the system can offer relevant and targeted suggestions to enhance their well-being.

Wireless Sensor Network (WSN) and Communication Protocols: The work highlights the importance of using WSNs and communication protocols to efficiently collect and transmit data from the sensors. The system uses wireless communication to transmit the collected data to a central hub or concentrator node, where it is processed and analyzed. The use of efficient communication protocols is crucial for optimizing data transmission and reducing energy consumption.

Energy-Saving Algorithm: The proposed energy-saving algorithm is integrated into the sensor system to optimize power consumption. It allows the system to be proactive or reactive, depending on the level of change in the measured parameters. Proactive mode ensures continuous monitoring when parameters show high variability, while reactive mode conserves energy by collecting data only on demand when the parameters are stable.

The impact of this work lies in its potential to revolutionize remote monitoring systems for elderly individuals, particularly in the context of sleep monitoring and healthcare assistance. By using WSNs and intelligent algorithms, the system can efficiently collect and analyze data, providing real-time feedback and personalized recommendations to improve sleep quality and overall health. The non-invasive nature of the system allows elderly individuals to receive care and assistance without the need for constant visits to healthcare centers, thus improving their quality of life and reducing the risk of health complications.

Comment 5: Spelling and grammar check is required.

Response: Many thanks to the Reviewer. We have revised and corrected the English style along the manuscript.

Thank you very much.

Sincerely,

Carolina Del-Valle-Soto

Universidad Panamericana. Facultad de Ingeniería. Álvaro del Portillo 49, Zapopan, Jalisco, 45010, México.

Phone: +52 (33) 13682200 | Ext. 4866

Reviewer 2 Report

Comments and Suggestions for Authors

The paper describes a substantial study of the use of a wireless sensor network used to monitor sleep patterns of older people and, based on those sleep patterns, recommend strategies to improve sleep. The paper presents a detailed statistical analysis of the sleep before and after. The key result is that sleep quality and duration is substantially improved as a result of the use of the system.

The results of the study are likely to be of interest to readers of the journal. The statistical analysis is thorough and convincing. The work is scientifically sound.

The research is of moderate significance. Monitoring of sleep and other physiological functions via short range networks has attracted a lot of interest over the past few years. The authors also include a description of an algorithm used for minimizing energy consumption of the network nodes.

Where there is a need for considerable work though is in the presentation of the results. There is much to be done to bring the manuscript up to an acceptable level. 

In particular the motivation section needs to be substantially revised to make very clear what the contribution is. The section needs to be explicit about what was done, what was measured, what interventions were carried out and what the results were. 

I found the detailed statistical analysis tedious to read. I will not insist on it but I think the authors should consider putting the detail in an appendix and just presenting the key findings in the body of the paper. 

The energy saving algorithm is not described at all well, nor is how it improves on other algorithms. The main metric is "energy increment" which we are informed is 3%. I found this mystifying. 3% of what? Later in a table on energy consumption there is a claim that the value is actually 50%. This section needs to be completely rewritten.

The "Discussion" Section seems to be just a repeat of the previous section. 

Comments on the Quality of English Language

The English needs quite a lot of work. Here are some things to begin with:

18 Keyword "robotics for healthcare" not appropriate.

25 "pertinent" is redundant

27 "Interesting" inappropriate. Could be "Improved healthcare monitoring becomes possible if..."

32 "this age group tends to have a high mortality rate" is stating the obvious. Older people tend to eventually die. The point being made is that if monitoring were improved then it might be possible to reduce mortality.

51 "This makes it possible to keep the person under control" is questionably expressed. Perhaps it means "monitor them more effectively".

65 Plural of "person" is (usually) "people"

67 When did "persons" (or people) become "patients"?

70 "and a high fraction of the population needs help in sleeping" A claim that needs justifying by a reference.

80 "metrics level of improvement." I do not know what this means.

90 "node utilization hierarchies to change the nature of the routing protocol." Not at all clear what this means.

94 "The results reveal that the network based on the sensors provides recommendations that improve sleep quality through a model of vital signs" is a very convoluted expression. Maybe "We found that recommendations based on measures from the sensors improved sleep quality".

97 Normally this section would end with an outline of what the rest of the paper contained.

164 The literature review is meant to show that there is a gap in the research work which this paper addresses. It is not a shopping list of what people have done in the area.

254 Table 4. It wasn't until I read this table that I understood what the paper was about. Some sort of summary describing these interventions needs to go in the Abstract and the Introduction. 

423 "grid resilience" is not explained.

553 Quite what the cluster analysis shows us is not at all clear.

Author Response

Dear

Editor

We are submitting the paper:

“Non-Invasive Monitoring of Vital Signs for the Elderly Using Low-Cost Wireless Sensor Networks: Exploring the Impact on Sleep and Home Security”

Authored by: Carolina Del-Valle-Soto *, Ramon A. Briseño, Leonardo J. Valdivia , Ramiro Velázquez , Juan Arturo Nolazco-Flores.

We would like to thank the reviewers and editors for their detailed analysis of the manuscript; the comments are very valuable to us. In the revised version of the paper, we have incorporated the all changes recommended by the reviewers.

Comments to all observations and suggestions including point-by-point responses are addressed in the following text.

Reviewer 2 comments

Comment 1: The paper describes a substantial study of the use of a wireless sensor network used to monitor sleep patterns of older people and, based on those sleep patterns, recommend strategies to improve sleep. The paper presents a detailed statistical analysis of the sleep before and after. The key result is that sleep quality and duration is substantially improved as a result of the use of the system.

The results of the study are likely to be of interest to readers of the journal. The statistical analysis is thorough and convincing. The work is scientifically sound.

The research is of moderate significance. Monitoring of sleep and other physiological functions via short range networks has attracted a lot of interest over the past few years. The authors also include a description of an algorithm used for minimizing energy consumption of the network nodes.

Where there is a need for considerable work though is in the presentation of the results. There is much to be done to bring the manuscript up to an acceptable level.

In particular the motivation section needs to be substantially revised to make very clear what the contribution is. The section needs to be explicit about what was done, what was measured, what interventions were carried out and what the results were.

Response: Many thanks to the Reviewer for his/her invaluable interest in the comments on this manuscript. We have added a paragraph in the Introduction that clarifies all the questions that the Reviewer properly raises.

The main contributions of this research work lie in developing and applying a wireless sensor network (WSN) for monitoring the vital signs of elderly individuals in their homes to improve their sleep quality. The environment is the same as the home of the elderly. The sensor network is installed in their home to be as strange or invasive as possible.

The study uses WSNs with low-cost sensors, including motion, pressure, temperature, humidity, noise, light, gyroscope, and air quality sensors, to gather real-time data from different house areas. The objective is to analyze the sleep metrics of older adults before and after implementing the sensor network to determine the impact of the system on improving rest. The proposed system aims to provide daily safety recommendations to the elderly and allow remote monitoring of vital signs, helping them to live independently and receive timely medical care. The research addresses the challenges of monitoring elderly individuals, especially in terms of their sleep patterns, as this age group tends to have reduced mobility and difficulty accessing health centers. The study demonstrates the potential of using WSNs and intelligent algorithms to optimize sensor utilization, improve sleep quality, and provide personalized recommendations for the elderly's well-being. The sensor system has a group of pre-written recommendations based on the possible causes of sensor parameter anomalies. These anomalies may be due to the person sleeping with much noise, getting up several times during the night, or the temperature in their home is too cold or too hot. They are only approximations of possible causes to improve the quality of life of people who live alone. This, being an approximation, is one of the system's limitations. The findings highlight the importance of continuous monitoring and non-invasive technology in enhancing the quality of life for older adults and preventing health complications. Overall, this work contributes to advancing remote monitoring systems for elderly individuals, particularly in the context of sleep monitoring and healthcare assistance.

Comment 2: I found the detailed statistical analysis tedious to read. I will not insist on it but I think the authors should consider putting the detail in an appendix and just presenting the key findings in the body of the paper.

Response: The Reviewer is correct, we have moved the mathematical support to an appendix at the end of the document (from page 30 to page 32) where the reader can consult if it is needed.

Comment 3: The energy saving algorithm is not described at all well, nor is how it improves on other algorithms. The main metric is "energy increment" which we are informed is 3%. I found this mystifying. 3% of what? Later in a table on energy consumption there is a claim that the value is actually 50%. This section needs to be completely rewritten.

Response: Thanks to Reviewer. We have improved the explanation related to the energy saving of the algorithm, to make the paper more readable.

Based on Algorithm 1, the system presents an energy-saving concerning the regular operation of the wireless sensor network. For this result, we consider the system's use for one hour, and when we apply the algorithm, we observe that the use of the sensor batteries is reduced. This is only an added value to the system because its principal value is monitoring the conditions of the house to give recommendations based on sleep.

We have also added this paragraph to the state of the art to complement the progress aspects of the proposed work.

The work presented here advances the state of the art in proposing a sensor system for monitoring people at home and providing personalized recommendations for improving sleep quality based on anomalous levels of measurements. This system leverages sensor technology and data analysis to offer solutions to enhance sleep patterns. The development of a sensor system tailored for home monitoring is a significant step forward. This system likely consists of various types of sensors strategically placed within the living spaces to capture relevant data related to sleep patterns. These sensors could include ambient light sensors, temperature sensors, motion sensors, heart rate monitors, and even advanced devices like sleep-specific wearable technology. The work proposes a data analysis framework to process the data collected by the sensor system effectively. By analyzing and interpreting the data, the system can identify anomalous levels or patterns in a person's sleep-related measurements. Anomalies could include irregular sleep duration, abnormal heart rate variations during sleep, or disturbances in sleep cycles. The key innovation lies in the system's ability to provide personalized recommendations based on the anomalous measurements. These recommendations could range from changes in sleep habits, lifestyle adjustments, relaxation techniques, or even suggestions to consult a medical professional, depending on the severity of the anomaly detected. The energy-saving algorithm integrated into the sensor system is another crucial advancement. By utilizing the programming nature of the sensors (proactive and reactive), the system can efficiently manage power consumption and extend the overall lifespan of the sensors. This algorithm likely employs techniques like dynamic sampling, adaptive data transmission, or intelligent sleep/wake-up modes for the sensors, ensuring that they are only active when necessary.

Comment 4: The "Discussion" Section seems to be just a repeat of the previous section.

Response: The Reviewer is correct, and we have refined the discussion section to avoid duplicating information with the preceding results section.

Comment 5: The English needs quite a lot of work. Here are some things to begin with:

18 Keyword "robotics for healthcare" not appropriate.

25 "pertinent" is redundant

27 "Interesting" inappropriate. Could be "Improved healthcare monitoring becomes possible if..."

32 "this age group tends to have a high mortality rate" is stating the obvious. Older people tend to eventually die. The point being made is that if monitoring were improved then it might be possible to reduce mortality.

51 "This makes it possible to keep the person under control" is questionably expressed. Perhaps it means "monitor them more effectively".

65 Plural of "person" is (usually) "people"

67 When did "persons" (or people) become "patients"?

70 "and a high fraction of the population needs help in sleeping" A claim that needs justifying by a reference.

80 "metrics level of improvement." I do not know what this means.

90 "node utilization hierarchies to change the nature of the routing protocol." Not at all clear what this means.

94 "The results reveal that the network based on the sensors provides recommendations that improve sleep quality through a model of vital signs" is a very convoluted expression. Maybe "We found that recommendations based on measures from the sensors improved sleep quality".

97 Normally this section would end with an outline of what the rest of the paper contained.

164 The literature review is meant to show that there is a gap in the research work which this paper addresses. It is not a shopping list of what people have done in the area.

254 Table 4. It wasn't until I read this table that I understood what the paper was about. Some sort of summary describing these interventions needs to go in the Abstract and the Introduction.

423 "grid resilience" is not explained.

553 Quite what the cluster analysis shows us is not at all clear.

Response: The Reviewer is correct. Many thanks to the Reviewer. We have revised and corrected the English style along the manuscript.

Moreover, regarding line 164, the Reviewer is right, and we have included a comparative table of representative works with advantages and limitations compared to our proposed system.

Table 2 provides a comprehensive comparison of various sleep monitoring mechanisms, including proposals from the literature and the state of the art. Each mechanism is evaluated based on its unique features and contributions. These existing mechanisms offer valuable insights into sleep monitoring, such as detecting sleep stages, assessing stress levels, monitoring sleep apnea, analyzing sleep patterns, observing sleep behavior, and tracking snoring. However, our proposed system, based on a non-invasive wireless sensor network (WSN) located in people's homes, introduces significant advancements in sleep monitoring technology. It operates without the need for wearables, making it more comfortable for users. The system monitors various parameters, such as luminosity, noise, and movement, providing a comprehensive view of the sleep environment and body parameters. By integrating multiple parameters, the proposed system can offer more accurate and personalized insights into sleep quality, generating personalized sleep recommendations. This non-wearable WSN in homes ensures a seamless and convenient monitoring experience, offering valuable improvements over existing mechanisms. Overall, the proposed system represents a promising and innovative approach to sleep monitoring, enhancing user comfort and providing actionable recommendations for better sleep quality.

Regarding line 254, the Reviewer's point is very coherent; therefore, we have added specific paragraphs in the Methodology, highlighting the contribution of these topics. This was detailed in the Introduction.

The connection between sensors, security, and the activities of the elderly is multifaceted:

Sensor System for Monitoring: The sensor system collects data from various sensors, including motion, pressure, temperature, humidity, noise, light, gyroscope, and air quality sensors. These sensors are designed to monitor the home environment and the elderly individual's vital signs while being unobtrusive and comfortable for the user.

Security and Privacy: The work emphasizes the need for non-invasiveness and respect for the privacy of the elderly. The data collected by the sensor network must be securely transmitted and stored to protect the individual's sensitive health information. Ensuring data security and privacy is crucial in healthcare applications, especially when dealing with sensitive data from vulnerable populations.

Activities of the Elderly: The sensor system aims to capture data related to the activities of the elderly in their homes. It tracks sleep patterns, movement, noise levels, and air quality to identify anomalies and provide personalized recommendations for improving sleep quality. By understanding the person's daily activities, the system can offer relevant and targeted suggestions to enhance their well-being.

Wireless Sensor Network (WSN) and Communication Protocols: The work highlights the importance of using WSNs and communication protocols to efficiently collect and transmit data from the sensors. The system uses wireless communication to transmit the collected data to a central hub or concentrator node, where it is processed and analyzed. The use of efficient communication protocols is crucial for optimizing data transmission and reducing energy consumption.

Energy-Saving Algorithm: The proposed energy-saving algorithm is integrated into the sensor system to optimize power consumption. It allows the system to be proactive or reactive, depending on the level of change in the measured parameters. Proactive mode ensures continuous monitoring when parameters show high variability, while reactive mode conserves energy by collecting data only on demand when the parameters are stable.

The impact of this work lies in its potential to revolutionize remote monitoring systems for elderly individuals, particularly in the context of sleep monitoring and healthcare assistance. By using WSNs and intelligent algorithms, the system can efficiently collect and analyze data, providing real-time feedback and personalized recommendations to improve sleep quality and overall health. The non-invasive nature of the system allows elderly individuals to receive care and assistance without the need for constant visits to healthcare centers, thus improving their quality of life and reducing the risk of health complications.

Regarding line 423, we have added:

Resilience WSNs refers to the network's ability to maintain its functionality and performance even in the face of various challenges, disturbances, or failures. These challenges can include node failures, communication link disruptions, environmental changes, or malicious attacks. A resilient WSN can adapt to changes and recover quickly from disruptions, ensuring the continuous and reliable operation of the network. The concept of resilience in WSNs is crucial because these networks are often deployed in dynamic and harsh environments where failures or disruptions are common. For example, in environmental monitoring applications, WSNs may be deployed in remote and inaccessible locations where nodes may fail due to harsh weather conditions or energy depletion. The ability of a network to return to its stable state after facing a disruption is known as "time to recovery" or "time to convergence." It measures the duration it takes for the network to recover its normal functioning and reach a stable state after a failure or disturbance. A shorter time to recovery indicates a more resilient network that can quickly adapt to changes and restore its operations.

Regarding line 553, we have refined the section dropping all the unnecessary cluster description information. Also, we clarified the goal of the cluster analysis in the methodology of the experiment section (3.2) and in the Cluster analysis section (4.2.3) to make the sections more comprehensible.

Also, we add this paragraph in the methodology of the experiment section:

This classification is important, as it was used to compare the metrics before and after utilizing the sensor network on sample groups with similar performance in their metrics. Furthermore, each sample was labeled based on its performance classification, and this label was used as the target variable in a multinomial multivariable logistic regression with a Logit parameter estimation (see in Appendix A section 7.9) performed to describe the predictive power of each metric in odds ratios and the relationships among them on the potential impact of the sensor network.

And this paragraph in the Cluster analysis section:

Labeling samples based on changes in metric performance is essential to perform regression analysis and discover predictive patterns in using the sensor network. Additionally, it allows analysis of differences between metrics before and after the use of the sensor network, for each group of samples isolated by label.

Thank you very much.

Sincerely,

Carolina Del-Valle-Soto

Universidad Panamericana. Facultad de Ingeniería. Álvaro del Portillo 49, Zapopan, Jalisco, 45010, México.

Phone: +52 (33) 13682200 | Ext. 4866

Reviewer 3 Report

Comments and Suggestions for Authors

The authors present a sensor network located in the home of an elderly person. The objective is to be able to non-invasively monitor the individual to give recommendations so that they have a better sleep. The paper explains too briefly the nodes of the sensor network. The state of the art of the technical part is reasonable, but the part closest to the field of health can be improved. The technical contribution of the authors is very little explained. The experiment carried out that affects people needs much more information: an elderly person without any medication is not the same as a person with a disease who may have sleep disorders due to it. English is good. Regarding the structure of the paper, there are some parts that could be eliminated and others that could be improved. The bibliography is acceptable.

Some possible improvements are detailed below.

1. Abstract. The sensor network does not improve sleep metrics, it will be the recommendations that are given.

2. Reference 2 is very technological, another one on medical sciences would be better here.

3. Before section 1.1, there is an abrupt end in the argument of the introduction. Something could be said about the sensor that detects sleep phases.

4. Line 82. What does independent person mean?

5. Line 85. This sentence is not understood. What will change in the routing?

6. Table 1. It is necessary to indicate how sleep has been measured. If any of the mentioned articles or a combination of them has been used. If a smart watch has been used all this could be removed and just describe the algorithm of the watch.

7. Table 2. How is the air quality sensor calibrated? These devices do not seem low cost, a solar panel is not cheap and surely it cannot be installed in all homes.

8. Table 3. Why have the indicated values been considered?

9. Line 240. what do you call a node? I don't understand what the LSI and RSSI have to do with it. If the nodes are fixed these values will not change. If a mobile sensor is considered, it will leave the coverage of one node to enter another, but it is difficult for it to access two at the same time.

10. Algorithm 1. It is a simple algorithm to control the activity of the node. It is also not understood what it is for, not understanding what we indicated in point 9.

11.Table 4. The table is our own or a scientific reference has been sought to support it. No typical message has been presented, no presentation has been made of how these recommendations are given based on the metrics.

12. Section 3.2 Is it 30 or 100 people? Data is missing such as the age of each person, mobility and/or diseases, how the nodes are located in the houses, the time of use, if there has been a problem, etc.

13. Line 278 -400. Most of it can be removed. The statistical development presented is not necessary.

14. Table5. It presents a very small improvement, it is not known how much it represents. The increase in energy over what is it? In which sensors has the energy consumption been measured?

15. Figure 2. I don't understand how resilience has been measured. It doesn't seem like much of a half second difference either.

16. Table 6. Since the structure of the nodes is not known, it is also not possible to know how the packets are routed. It is also not known how the experimentation to indicate an interval has been done. Have measurements been made for two weeks for each row? A lot of data is missing to begin to understand this table.

17. Figure 3. What or who gives these alerts?

18. Table 7. How have these metrics been measured? Could the table be subdivided for each of the sleep phases?

19. Table 7. When it is indicated in the table that the sensor network is not used, what it means is that no recommendations are made.

20. Table 7. How many recommendations have been made?

21. Section 4.2.3 I don't see much use in it. Knowing who has benefited and who hasn't would be enough.

22. Section 5. This section explains relatively well how the system could be used and how new lines of action could be incorporated. It should be shortened.

Comments on the Quality of English Language

English sounds well. 

Author Response

Dear

Editor

We are submitting the paper:

“Non-Invasive Monitoring of Vital Signs for the Elderly Using Low-Cost Wireless Sensor Networks: Exploring the Impact on Sleep and Home Security”

Authored by: Carolina Del-Valle-Soto *, Ramon A. Briseño, Leonardo J. Valdivia , Ramiro Velázquez , Juan Arturo Nolazco-Flores.

We would like to thank the reviewers and editors for their detailed analysis of the manuscript; the comments are very valuable to us. In the revised version of the paper, we have incorporated the all changes recommended by the reviewers.

Comments to all observations and suggestions including point-by-point responses are addressed in the following text.

Reviewer 1 comments

Comment 1: The main contributions of this work are not clearly highlighted. The paper should clearly give the environment, assumptions, requirements and objectives of the problem in hand, and points out major issues or difficulties when dealing with the problem.

Response: Many thanks to the Reviewer for his/her invaluable interest in the comments on this manuscript. We have added a paragraph in the Introduction that clarifies all the questions that the Reviewer properly raises.

The main contributions of this research work lie in developing and applying a wireless sensor network (WSN) for monitoring the vital signs of elderly individuals in their homes to improve their sleep quality. The environment is the same as the home of the elderly. The sensor network is installed in their home to be as strange or invasive as possible.

The study uses WSNs with low-cost sensors, including motion, pressure, temperature, humidity, noise, light, gyroscope, and air quality sensors, to gather real-time data from different house areas. The objective is to analyze the sleep metrics of older adults before and after implementing the sensor network to determine the impact of the system on improving rest. The proposed system aims to provide daily safety recommendations to the elderly and allow remote monitoring of vital signs, helping them to live independently and receive timely medical care. The research addresses the challenges of monitoring elderly individuals, especially in terms of their sleep patterns, as this age group tends to have reduced mobility and difficulty accessing health centers. The study demonstrates the potential of using WSNs and intelligent algorithms to optimize sensor utilization, improve sleep quality, and provide personalized recommendations for the elderly's well-being. The sensor system has a group of pre-written recommendations based on the possible causes of sensor parameter anomalies. These anomalies may be due to the person sleeping with much noise, getting up several times during the night, or the temperature in their home is too cold or too hot. They are only approximations of possible causes to improve the quality of life of people who live alone. This, being an approximation, is one of the system's limitations. The findings highlight the importance of continuous monitoring and non-invasive technology in enhancing the quality of life for older adults and preventing health complications. Overall, this work contributes to advancing remote monitoring systems for elderly individuals, particularly in the context of sleep monitoring and healthcare assistance.

Comment 2: Vest amounts of existing studies have proposed the similar concept to monitor sleep behaviors using the WSNs. This paper should review the class of these papers and clearly give the differences and major improvements of the proposed mechanism. Thus, the literature gap filled by the present study is not clearly stated. Please expand on this statement.

Response: Many thanks to the Reviewer. The Reviewer is right, and we have included a comparative table of representative works with advantages and limitations compared to our proposed system.

Table 2 provides a comprehensive comparison of various sleep monitoring mechanisms, including proposals from the literature and the state of the art. Each mechanism is evaluated based on its unique features and contributions. These existing mechanisms offer valuable insights into sleep monitoring, such as detecting sleep stages, assessing stress levels, monitoring sleep apnea, analyzing sleep patterns, observing sleep behavior, and tracking snoring. However, our proposed system, based on a non-invasive wireless sensor network (WSN) located in people's homes, introduces significant advancements in sleep monitoring technology. It operates without the need for wearables, making it more comfortable for users. The system monitors various parameters, such as luminosity, noise, and movement, providing a comprehensive view of the sleep environment and body parameters. By integrating multiple parameters, the proposed system can offer more accurate and personalized insights into sleep quality, generating personalized sleep recommendations. This non-wearable WSN in homes ensures a seamless and convenient monitoring experience, offering valuable improvements over existing mechanisms. Overall, the proposed system represents a promising and innovative approach to sleep monitoring, enhancing user comfort and providing actionable recommendations for better sleep quality.

Comment 3: The discussion and evaluation part (Section IV) only considers several different parameter settings for the authors’ solutions and compares this work to the conventional schemes (i.e., with/without WSNs and with/without energy saving alg.). Therefore, it is unclear if and how this work advances the state of the art. Please expand on this statement.

Response: Thanks to Reviewer. We have improved the explanation related to the energy saving of the algorithm, to make the paper more readable.

Based on Algorithm 1, the system presents an energy-saving concerning the regular operation of the wireless sensor network. For this result, we consider the system's use for one hour, and when we apply the algorithm, we observe that the use of the sensor batteries is reduced. This is only an added value to the system because its principal value is monitoring the conditions of the house to give recommendations based on sleep.

We have also added this paragraph to the state of the art to complement the progress aspects of the proposed work.

The work presented here advances the state of the art in proposing a sensor system for monitoring people at home and providing personalized recommendations for improving sleep quality based on anomalous levels of measurements. This system leverages sensor technology and data analysis to offer solutions to enhance sleep patterns. The development of a sensor system tailored for home monitoring is a significant step forward. This system likely consists of various types of sensors strategically placed within the living spaces to capture relevant data related to sleep patterns. These sensors could include ambient light sensors, temperature sensors, motion sensors, heart rate monitors, and even advanced devices like sleep-specific wearable technology. The work proposes a data analysis framework to process the data collected by the sensor system effectively. By analyzing and interpreting the data, the system can identify anomalous levels or patterns in a person's sleep-related measurements. Anomalies could include irregular sleep duration, abnormal heart rate variations during sleep, or disturbances in sleep cycles. The key innovation lies in the system's ability to provide personalized recommendations based on the anomalous measurements. These recommendations could range from changes in sleep habits, lifestyle adjustments, relaxation techniques, or even suggestions to consult a medical professional, depending on the severity of the anomaly detected. The energy-saving algorithm integrated into the sensor system is another crucial advancement. By utilizing the programming nature of the sensors (proactive and reactive), the system can efficiently manage power consumption and extend the overall lifespan of the sensors. This algorithm likely employs techniques like dynamic sampling, adaptive data transmission, or intelligent sleep/wake-up modes for the sensors, ensuring that they are only active when necessary.

Comment 4: The connection among the sensors, security, the activities of the elder population is too weak. Please expand on this statement. 

Response: The Reviewer's point is very coherent; therefore, we have added specific paragraphs in the Methodology, highlighting the contribution of these topics.

The connection between sensors, security, and the activities of the elderly is multifaceted:

Sensor System for Monitoring: The sensor system collects data from various sensors, including motion, pressure, temperature, humidity, noise, light, gyroscope, and air quality sensors. These sensors are designed to monitor the home environment and the elderly individual's vital signs while being unobtrusive and comfortable for the user.

Security and Privacy: The work emphasizes the need for non-invasiveness and respect for the privacy of the elderly. The data collected by the sensor network must be securely transmitted and stored to protect the individual's sensitive health information. Ensuring data security and privacy is crucial in healthcare applications, especially when dealing with sensitive data from vulnerable populations.

Activities of the Elderly: The sensor system aims to capture data related to the activities of the elderly in their homes. It tracks sleep patterns, movement, noise levels, and air quality to identify anomalies and provide personalized recommendations for improving sleep quality. By understanding the person's daily activities, the system can offer relevant and targeted suggestions to enhance their well-being.

Wireless Sensor Network (WSN) and Communication Protocols: The work highlights the importance of using WSNs and communication protocols to efficiently collect and transmit data from the sensors. The system uses wireless communication to transmit the collected data to a central hub or concentrator node, where it is processed and analyzed. The use of efficient communication protocols is crucial for optimizing data transmission and reducing energy consumption.

Energy-Saving Algorithm: The proposed energy-saving algorithm is integrated into the sensor system to optimize power consumption. It allows the system to be proactive or reactive, depending on the level of change in the measured parameters. Proactive mode ensures continuous monitoring when parameters show high variability, while reactive mode conserves energy by collecting data only on demand when the parameters are stable.

The impact of this work lies in its potential to revolutionize remote monitoring systems for elderly individuals, particularly in the context of sleep monitoring and healthcare assistance. By using WSNs and intelligent algorithms, the system can efficiently collect and analyze data, providing real-time feedback and personalized recommendations to improve sleep quality and overall health. The non-invasive nature of the system allows elderly individuals to receive care and assistance without the need for constant visits to healthcare centers, thus improving their quality of life and reducing the risk of health complications.

Comment 5: Spelling and grammar check is required.

Response: Many thanks to the Reviewer. We have revised and corrected the English style along the manuscript.

Reviewer 2 comments

Comment 1: The paper describes a substantial study of the use of a wireless sensor network used to monitor sleep patterns of older people and, based on those sleep patterns, recommend strategies to improve sleep. The paper presents a detailed statistical analysis of the sleep before and after. The key result is that sleep quality and duration is substantially improved as a result of the use of the system.

The results of the study are likely to be of interest to readers of the journal. The statistical analysis is thorough and convincing. The work is scientifically sound.

The research is of moderate significance. Monitoring of sleep and other physiological functions via short range networks has attracted a lot of interest over the past few years. The authors also include a description of an algorithm used for minimizing energy consumption of the network nodes.

Where there is a need for considerable work though is in the presentation of the results. There is much to be done to bring the manuscript up to an acceptable level.

In particular the motivation section needs to be substantially revised to make very clear what the contribution is. The section needs to be explicit about what was done, what was measured, what interventions were carried out and what the results were.

Response: Many thanks to the Reviewer for his/her invaluable interest in the comments on this manuscript. We have added a paragraph in the Introduction that clarifies all the questions that the Reviewer properly raises.

The main contributions of this research work lie in developing and applying a wireless sensor network (WSN) for monitoring the vital signs of elderly individuals in their homes to improve their sleep quality. The environment is the same as the home of the elderly. The sensor network is installed in their home to be as strange or invasive as possible.

The study uses WSNs with low-cost sensors, including motion, pressure, temperature, humidity, noise, light, gyroscope, and air quality sensors, to gather real-time data from different house areas. The objective is to analyze the sleep metrics of older adults before and after implementing the sensor network to determine the impact of the system on improving rest. The proposed system aims to provide daily safety recommendations to the elderly and allow remote monitoring of vital signs, helping them to live independently and receive timely medical care. The research addresses the challenges of monitoring elderly individuals, especially in terms of their sleep patterns, as this age group tends to have reduced mobility and difficulty accessing health centers. The study demonstrates the potential of using WSNs and intelligent algorithms to optimize sensor utilization, improve sleep quality, and provide personalized recommendations for the elderly's well-being. The sensor system has a group of pre-written recommendations based on the possible causes of sensor parameter anomalies. These anomalies may be due to the person sleeping with much noise, getting up several times during the night, or the temperature in their home is too cold or too hot. They are only approximations of possible causes to improve the quality of life of people who live alone. This, being an approximation, is one of the system's limitations. The findings highlight the importance of continuous monitoring and non-invasive technology in enhancing the quality of life for older adults and preventing health complications. Overall, this work contributes to advancing remote monitoring systems for elderly individuals, particularly in the context of sleep monitoring and healthcare assistance.

Comment 2: I found the detailed statistical analysis tedious to read. I will not insist on it but I think the authors should consider putting the detail in an appendix and just presenting the key findings in the body of the paper.

Response: The Reviewer is correct, we have moved the mathematical support to an appendix at the end of the document (from page 30 to page 32) where the reader can consult if it is needed.

Comment 3: The energy saving algorithm is not described at all well, nor is how it improves on other algorithms. The main metric is "energy increment" which we are informed is 3%. I found this mystifying. 3% of what? Later in a table on energy consumption there is a claim that the value is actually 50%. This section needs to be completely rewritten.

Response: Thanks to Reviewer. We have improved the explanation related to the energy saving of the algorithm, to make the paper more readable.

Based on Algorithm 1, the system presents an energy-saving concerning the regular operation of the wireless sensor network. For this result, we consider the system's use for one hour, and when we apply the algorithm, we observe that the use of the sensor batteries is reduced. This is only an added value to the system because its principal value is monitoring the conditions of the house to give recommendations based on sleep.

We have also added this paragraph to the state of the art to complement the progress aspects of the proposed work.

The work presented here advances the state of the art in proposing a sensor system for monitoring people at home and providing personalized recommendations for improving sleep quality based on anomalous levels of measurements. This system leverages sensor technology and data analysis to offer solutions to enhance sleep patterns. The development of a sensor system tailored for home monitoring is a significant step forward. This system likely consists of various types of sensors strategically placed within the living spaces to capture relevant data related to sleep patterns. These sensors could include ambient light sensors, temperature sensors, motion sensors, heart rate monitors, and even advanced devices like sleep-specific wearable technology. The work proposes a data analysis framework to process the data collected by the sensor system effectively. By analyzing and interpreting the data, the system can identify anomalous levels or patterns in a person's sleep-related measurements. Anomalies could include irregular sleep duration, abnormal heart rate variations during sleep, or disturbances in sleep cycles. The key innovation lies in the system's ability to provide personalized recommendations based on the anomalous measurements. These recommendations could range from changes in sleep habits, lifestyle adjustments, relaxation techniques, or even suggestions to consult a medical professional, depending on the severity of the anomaly detected. The energy-saving algorithm integrated into the sensor system is another crucial advancement. By utilizing the programming nature of the sensors (proactive and reactive), the system can efficiently manage power consumption and extend the overall lifespan of the sensors. This algorithm likely employs techniques like dynamic sampling, adaptive data transmission, or intelligent sleep/wake-up modes for the sensors, ensuring that they are only active when necessary.

Comment 4: The "Discussion" Section seems to be just a repeat of the previous section.

Response: The Reviewer is correct, and we have refined the discussion section to avoid duplicating information with the preceding results section.

Comment 5: The English needs quite a lot of work. Here are some things to begin with:

18 Keyword "robotics for healthcare" not appropriate.

25 "pertinent" is redundant

27 "Interesting" inappropriate. Could be "Improved healthcare monitoring becomes possible if..."

32 "this age group tends to have a high mortality rate" is stating the obvious. Older people tend to eventually die. The point being made is that if monitoring were improved then it might be possible to reduce mortality.

51 "This makes it possible to keep the person under control" is questionably expressed. Perhaps it means "monitor them more effectively".

65 Plural of "person" is (usually) "people"

67 When did "persons" (or people) become "patients"?

70 "and a high fraction of the population needs help in sleeping" A claim that needs justifying by a reference.

80 "metrics level of improvement." I do not know what this means.

90 "node utilization hierarchies to change the nature of the routing protocol." Not at all clear what this means.

94 "The results reveal that the network based on the sensors provides recommendations that improve sleep quality through a model of vital signs" is a very convoluted expression. Maybe "We found that recommendations based on measures from the sensors improved sleep quality".

97 Normally this section would end with an outline of what the rest of the paper contained.

164 The literature review is meant to show that there is a gap in the research work which this paper addresses. It is not a shopping list of what people have done in the area.

254 Table 4. It wasn't until I read this table that I understood what the paper was about. Some sort of summary describing these interventions needs to go in the Abstract and the Introduction.

423 "grid resilience" is not explained.

553 Quite what the cluster analysis shows us is not at all clear.

Response: The Reviewer is correct. Many thanks to the Reviewer. We have revised and corrected the English style along the manuscript.

Moreover, regarding line 164, the Reviewer is right, and we have included a comparative table of representative works with advantages and limitations compared to our proposed system.

Table 2 provides a comprehensive comparison of various sleep monitoring mechanisms, including proposals from the literature and the state of the art. Each mechanism is evaluated based on its unique features and contributions. These existing mechanisms offer valuable insights into sleep monitoring, such as detecting sleep stages, assessing stress levels, monitoring sleep apnea, analyzing sleep patterns, observing sleep behavior, and tracking snoring. However, our proposed system, based on a non-invasive wireless sensor network (WSN) located in people's homes, introduces significant advancements in sleep monitoring technology. It operates without the need for wearables, making it more comfortable for users. The system monitors various parameters, such as luminosity, noise, and movement, providing a comprehensive view of the sleep environment and body parameters. By integrating multiple parameters, the proposed system can offer more accurate and personalized insights into sleep quality, generating personalized sleep recommendations. This non-wearable WSN in homes ensures a seamless and convenient monitoring experience, offering valuable improvements over existing mechanisms. Overall, the proposed system represents a promising and innovative approach to sleep monitoring, enhancing user comfort and providing actionable recommendations for better sleep quality.

Regarding line 254, the Reviewer's point is very coherent; therefore, we have added specific paragraphs in the Methodology, highlighting the contribution of these topics. This was detailed in the Introduction.

The connection between sensors, security, and the activities of the elderly is multifaceted:

Sensor System for Monitoring: The sensor system collects data from various sensors, including motion, pressure, temperature, humidity, noise, light, gyroscope, and air quality sensors. These sensors are designed to monitor the home environment and the elderly individual's vital signs while being unobtrusive and comfortable for the user.

Security and Privacy: The work emphasizes the need for non-invasiveness and respect for the privacy of the elderly. The data collected by the sensor network must be securely transmitted and stored to protect the individual's sensitive health information. Ensuring data security and privacy is crucial in healthcare applications, especially when dealing with sensitive data from vulnerable populations.

Activities of the Elderly: The sensor system aims to capture data related to the activities of the elderly in their homes. It tracks sleep patterns, movement, noise levels, and air quality to identify anomalies and provide personalized recommendations for improving sleep quality. By understanding the person's daily activities, the system can offer relevant and targeted suggestions to enhance their well-being.

Wireless Sensor Network (WSN) and Communication Protocols: The work highlights the importance of using WSNs and communication protocols to efficiently collect and transmit data from the sensors. The system uses wireless communication to transmit the collected data to a central hub or concentrator node, where it is processed and analyzed. The use of efficient communication protocols is crucial for optimizing data transmission and reducing energy consumption.

Energy-Saving Algorithm: The proposed energy-saving algorithm is integrated into the sensor system to optimize power consumption. It allows the system to be proactive or reactive, depending on the level of change in the measured parameters. Proactive mode ensures continuous monitoring when parameters show high variability, while reactive mode conserves energy by collecting data only on demand when the parameters are stable.

The impact of this work lies in its potential to revolutionize remote monitoring systems for elderly individuals, particularly in the context of sleep monitoring and healthcare assistance. By using WSNs and intelligent algorithms, the system can efficiently collect and analyze data, providing real-time feedback and personalized recommendations to improve sleep quality and overall health. The non-invasive nature of the system allows elderly individuals to receive care and assistance without the need for constant visits to healthcare centers, thus improving their quality of life and reducing the risk of health complications.

Regarding line 423, we have added:

Resilience WSNs refers to the network's ability to maintain its functionality and performance even in the face of various challenges, disturbances, or failures. These challenges can include node failures, communication link disruptions, environmental changes, or malicious attacks. A resilient WSN can adapt to changes and recover quickly from disruptions, ensuring the continuous and reliable operation of the network. The concept of resilience in WSNs is crucial because these networks are often deployed in dynamic and harsh environments where failures or disruptions are common. For example, in environmental monitoring applications, WSNs may be deployed in remote and inaccessible locations where nodes may fail due to harsh weather conditions or energy depletion. The ability of a network to return to its stable state after facing a disruption is known as "time to recovery" or "time to convergence." It measures the duration it takes for the network to recover its normal functioning and reach a stable state after a failure or disturbance. A shorter time to recovery indicates a more resilient network that can quickly adapt to changes and restore its operations.

Regarding line 553, we have refined the section dropping all the unnecessary cluster description information. Also, we clarified the goal of the cluster analysis in the methodology of the experiment section (3.2) and in the Cluster analysis section (4.2.3) to make the sections more comprehensible.

Also, we add this paragraph in the methodology of the experiment section:

This classification is important, as it was used to compare the metrics before and after utilizing the sensor network on sample groups with similar performance in their metrics. Furthermore, each sample was labeled based on its performance classification, and this label was used as the target variable in a multinomial multivariable logistic regression with a Logit parameter estimation (see in Appendix A section 7.9) performed to describe the predictive power of each metric in odds ratios and the relationships among them on the potential impact of the sensor network.

And this paragraph in the Cluster analysis section:

Labeling samples based on changes in metric performance is essential to perform regression analysis and discover predictive patterns in using the sensor network. Additionally, it allows analysis of differences between metrics before and after the use of the sensor network, for each group of samples isolated by label.

Reviewer 3 comments

Comment 1: The authors present a sensor network located in the home of an elderly person. The objective is to be able to non-invasively monitor the individual to give recommendations so that they have a better sleep. The paper explains too briefly the nodes of the sensor network. The state of the art of the technical part is reasonable, but the part closest to the field of health can be improved. The technical contribution of the authors is very little explained. The experiment carried out that affects people needs much more information: an elderly person without any medication is not the same as a person with a disease who may have sleep disorders due to it. English is good. Regarding the structure of the paper, there are some parts that could be eliminated and others that could be improved. The bibliography is acceptable.

Some possible improvements are detailed below.

  1. Abstract. The sensor network does not improve sleep metrics, it will be the recommendations that are given.

Response: Thanks to Reviewer. We have changed this wording in the abstract so that the meaning of the sentence is not lost.

Comment 2: Reference 2 is very technological, another one on medical sciences would be better here.

Response: The Reviewer is correct and we have changed the reference as appropriately the Reviewer proposes.

Mansouri, S. The Development of the Vital Signs Tele-monitoring System for the Elderly by 935

Using UML Language and the Interoperability Standard Continua. The Open Bioinformatics 936

Journal 2020, 13.

Comment 3: Before section 1.1, there is an abrupt end in the argument of the introduction. Something could be said about the sensor that detects sleep phases.

Response: Many thanks to the Reviewer for his/her invaluable interest in the comments on this manuscript. We have added a paragraph in the Introduction that clarifies all the questions that the Reviewer properly raises.

The main contributions of this research work lie in developing and applying a wireless sensor network (WSN) for monitoring the vital signs of elderly individuals in their homes to improve their sleep quality. The environment is the same as the home of the elderly. The sensor network is installed in their home to be as strange or invasive as possible.

The study uses WSNs with low-cost sensors, including motion, pressure, temperature, humidity, noise, light, gyroscope, and air quality sensors, to gather real-time data from different house areas. The objective is to analyze the sleep metrics of older adults before and after implementing the sensor network to determine the impact of the system on improving rest. The proposed system aims to provide daily safety recommendations to the elderly and allow remote monitoring of vital signs, helping them to live independently and receive timely medical care. The research addresses the challenges of monitoring elderly individuals, especially in terms of their sleep patterns, as this age group tends to have reduced mobility and difficulty accessing health centers. The study demonstrates the potential of using WSNs and intelligent algorithms to optimize sensor utilization, improve sleep quality, and provide personalized recommendations for the elderly's well-being. The sensor system has a group of pre-written recommendations based on the possible causes of sensor parameter anomalies. These anomalies may be due to the person sleeping with much noise, getting up several times during the night, or the temperature in their home is too cold or too hot. They are only approximations of possible causes to improve the quality of life of people who live alone. This, being an approximation, is one of the system's limitations. The findings highlight the importance of continuous monitoring and non-invasive technology in enhancing the quality of life for older adults and preventing health complications. Overall, this work contributes to advancing remote monitoring systems for elderly individuals, particularly in the context of sleep monitoring and healthcare assistance.

Comment 4: Line 82. What does independent person mean?

Response: Thanks to Reviewer and we've added a paragraph that naturally connects both ideas.

In the context of the provided passage, an "independent person" refers to an elderly individual who wishes to live on their own and maintain their autonomy without relying heavily on constant assistance or care from others. This could be an elderly person who prefers to live in their own home rather than moving to an assisted living facility or nursing home.

Comment 5: Line 85. This sentence is not understood. What will change in the routing?

Response: Thanks to the Reviewer for his observation. We have clarified this explanation regarding the behavior of the routing protocol:

This system monitors the main areas of the house programmed with a low energy cost algorithm and sends recommendations to the person at the beginning of each day. It is a simple algorithm based on based on the threshold of measurements perceived by the sensors during the day. In this way, the nodes change hierarchies in the network and have proactive or reactive priorities for sending information. The routing protocol takes advantage of each node's proactive or reactive nature to transmit the packets. The change that is optimized in the proposed algorithm is that when the node's measurement levels are kept within a normal range, the protocol responds to sending packets reactively. Whereas if the measurement levels are outside the threshold, the protocol responds proactively to sending packets to better control the information review.

Comment 6: Table 1. It is necessary to indicate how sleep has been measured. If any of the mentioned articles or a combination of them has been used. If a smart watch has been used all this could be removed and just describe the algorithm of the watch.

Response: Table 1 only compares works related to sleep measurements and vital signs of people to keep track of them. Not in all cases a smartwatch is used (as it is in our case because we have other equally crucial vital sign metrics). With the permission of the Reviewer, we have considered keeping this Table so that the reader has an overview of the state of the art with notable parameters and characteristics.

Comment 7: Table 2. How is the air quality sensor calibrated? These devices do not seem low cost, a solar panel is not cheap and surely it cannot be installed in all homes.

Response: Many thanks to the Reviewer. We understand the Reviewer's concern, and it is very accurate. However, we have purchased these sensors in an internal university project, and the cost has been cheaper for wholesale purchase. The sensors can be powered by a solar cell or by current in a plug-and-play way. We showed the Reviewer a couple of photos so they could see the different configurations and packaging we had for the sensors so they could be used inside or outside the home. The value of this sensor is approximately 35 USD.

Comment 8: Table 3. Why have the indicated values been considered?

Response: Thanks to the Reviewer for his/her pertinent comment. Values are approximate average ranges in medicine. We have added a reference to formalize the data.

Saul, L. (2006). The effect of repeat exercise on exercise-induced arterial hypoxemia.

Comment 9: Line 240. what do you call a node? I don't understand what the LSI and RSSI have to do with it. If the nodes are fixed these values will not change. If a mobile sensor is considered, it will leave the coverage of one node to enter another, but it is difficult for it to access two at the same time.

Response: We understand the Reviewer's concern. The node is the device that has or can have multiple sensors. Indeed, the RSSI and LQI values should not change in static nodes. However, this can be altered by other devices that the person activates one day or by environmental conditions. Furthermore, when these factors change, there are anomalous environmental conditions, precisely what we want to analyze to establish the recommendations.

This paragraph was added there to make the intent of the algorithm's measurement clearer.

Comment 10: Algorithm 1. It is a simple algorithm to control the activity of the node. It is also not understood what it is for, not understanding what we indicated in point 9.

Response: We agree with the Reviewer. Precisely, we present a simple algorithm that modifies the nature of packet delivery (proactive or reactive) according to the variation thresholds of the sensor measurement. We intend to make something other than a fancy algorithm to make it energy efficient.

We have also added this paragraph:

Energy-Saving Algorithm: The proposed energy-saving algorithm is integrated into the sensor system to optimize power consumption. It allows the system to be proactive or reactive, depending on the level of change in the measured parameters. Proactive mode ensures continuous monitoring when parameters show high variability, while reactive mode conserves energy by collecting data only on demand when the parameters are stable.

Comment 11: Table 4. The table is our own or a scientific reference has been sought to support it. No typical message has been presented, no presentation has been made of how these recommendations are given based on the metrics.

Response: We agree with the Reviewer. The Table is self-made. This Table has recommendations adapted to the general functioning of an average house in Mexico in an urban area. Each one of the groups is related to characteristics of the house or of the people, which could directly or indirectly impact the way the person sleeps. For example, Group 1 presents recommendations for the house or its composition. Group 2 presents recommendations mainly related to the person, which can be part of their daily life and habits. Moreover, Group 3 has recommendations about the person's bedroom.

Table 5 provides recommendations for sleeping better at home, grouped into three different categories: Group 1, Group 2, and Group 3. Each group contains a list of seven different recommendations aimed at helping individuals improve their sleep quality. Group 1 recommendations focus on creating a comfortable and relaxing sleep environment. This includes closing the window to block out noise, using a comfortable mattress and pillows, using a noise machine or earplugs to block out background noise, keeping the room at a cool temperature, using a humidifier to improve air quality, keeping the room dark and using black-out curtains, and avoiding caffeine and heavy meals before bedtime. All these recommendations are aimed at creating a sleep-conducive environment that is comfortable and free from distractions. Group 2 recommendations focus on relaxation and preparation for sleep. These recommendations include practicing relaxation techniques before bedtime, keeping a consistent sleep schedule, using a weighted blanket, trying aromatherapy with essential oils, keeping electronics out of the bedroom, using a white noise machine, and trying a sleep mask. All these recommendations are aimed at relaxing the body and mind and preparing them for sleep. Group 3 recommendations focus on developing good sleep habits, and include establishing a bedtime routine, exercising regularly but not close to bedtime, avoiding blue light exposure before bedtime, considering using a sleep aid, avoiding napping during the day, trying a natural sleep remedy like melatonin, making sure your bed is comfortable and supportive. All these recommendations are aimed at developing good sleep habits which are conducive for a good night’s sleep.

Comment 12: Section 3.2 Is it 30 or 100 people? Data is missing such as the age of each person, mobility and/or diseases, how the nodes are located in the houses, the time of use, if there has been a problem, etc.

Response: Thanks to Reviewer. We have completed this subsection:

The age of the people is between 50 and 70 years. There are 16 women and 14 men. People are relatively healthy concerning their health. They do not have heart or respiratory diseases (because this could alter the results a bit).  The collected data will likely provide valuable insights into sleep patterns and overall well-being for this demographic. A consent to approve the use of the system and its installation in their home is signed. The nodes adapt according to the layout of each person's house. There must be at least two nodes in the room where the person sleeps, and in the main areas of the house (kitchen, living room, and dining room), at least one node. The nodes are easy to install near a power outlet. The nodes' adaptability based on the layout of each person's home is crucial. This ensures that the sensor system can effectively capture relevant data points while being minimally invasive and seamlessly integrated into the participants' living spaces.

Comment 13: Line 278 -400. Most of it can be removed. The statistical development presented is not necessary.

Response: The reviewer's perspective is valid; the presented statistical information may not be necessary. Therefore, we have relocated this information to an appendix at the end of the article (from page 30 to 32).

Comment 14: Table5. It presents a very small improvement, it is not known how much it represents. The increase in energy over what is it? In which sensors has the energy consumption been measured?

Response: Thanks to Reviewer. We have improved the explanation related to the energy saving of the algorithm, to make the paper more readable.

Based on Algorithm 1, the system presents an energy-saving concerning the regular operation of the wireless sensor network. For this result, we consider the system's use for one hour, and when we apply the algorithm, we observe that the use of the sensor batteries is reduced. This is only an added value to the system because its principal value is monitoring the conditions of the house to give recommendations based on sleep.

We have also added this paragraph to the state of the art to complement the progress aspects of the proposed work.

The work presented here advances the state of the art in proposing a sensor system for monitoring people at home and providing personalized recommendations for improving sleep quality based on anomalous levels of measurements. This system leverages sensor technology and data analysis to offer solutions to enhance sleep patterns. The development of a sensor system tailored for home monitoring is a significant step forward. This system likely consists of various types of sensors strategically placed within the living spaces to capture relevant data related to sleep patterns. These sensors could include ambient light sensors, temperature sensors, motion sensors, heart rate monitors, and even advanced devices like sleep-specific wearable technology. The work proposes a data analysis framework to process the data collected by the sensor system effectively. By analyzing and interpreting the data, the system can identify anomalous levels or patterns in a person's sleep-related measurements. Anomalies could include irregular sleep duration, abnormal heart rate variations during sleep, or disturbances in sleep cycles. The key innovation lies in the system's ability to provide personalized recommendations based on the anomalous measurements. These recommendations could range from changes in sleep habits, lifestyle adjustments, relaxation techniques, or even suggestions to consult a medical professional, depending on the severity of the anomaly detected. The energy-saving algorithm integrated into the sensor system is another crucial advancement. By utilizing the programming nature of the sensors (proactive and reactive), the system can efficiently manage power consumption and extend the overall lifespan of the sensors. This algorithm likely employs techniques like dynamic sampling, adaptive data transmission, or intelligent sleep/wake-up modes for the sensors, ensuring that they are only active when necessary.

Comment 15: Figure 2. I don't understand how resilience has been measured. It doesn't seem like much of a half second difference either.

Response: We thank the Reviewer. Resilience WSNs refers to the network's ability to maintain its functionality and performance even in the face of various challenges, disturbances, or failures. These challenges can include node failures, communication link disruptions, environmental changes, or malicious attacks. A resilient WSN can adapt to changes and recover quickly from disruptions, ensuring the continuous and reliable operation of the network. The concept of resilience in WSNs is crucial because these networks are often deployed in dynamic and harsh environments where failures or disruptions are common. For example, in environmental monitoring applications, WSNs may be deployed in remote and inaccessible locations where nodes may fail due to harsh weather conditions or energy depletion. The ability of a network to return to its stable state after facing a disruption is known as "time to recovery" or "time to convergence." It measures the duration it takes for the network to recover its normal functioning and reach a stable state after a failure or disturbance. A shorter time to recovery indicates a more resilient network that can quickly adapt to changes and restore its operations.

Comment 16: Table 6. Since the structure of the nodes is not known, it is also not possible to know how the packets are routed. It is also not known how the experimentation to indicate an interval has been done. Have measurements been made for two weeks for each row? A lot of data is missing to begin to understand this table.

Response: We have corrected this Table. We understand the Reviewer's concern and have relocated this Table because it is state-of-the-art and not a table of our algorithm. The Table is now in the Related Work section.

Comment 17: Figure 3. What or who gives these alerts?

Response: Thanks to the Reviewer. The Reviewer is right. That Figure needed to be clarified and explained. We have added and supplemented the following information:

The sensors send the alerts via email or SMS to the person to show him if a sensor presents more significant variation than usual. This is so that the person knows that behavior is not within the established thresholds in that area of the house. For example, a more significant variation in air quality, higher abnormal noise levels, or greater use of particular doors at night means that the person is not resting properly, etc. The purpose of evidencing these alerts from the sensor system is that they should decrease as the system provides the person with recommendations daily.

Comment 18: Table 7. How have these metrics been measured? Could the table be subdivided for each of the sleep phases?

Response: We have corrected this naming error.

Comment 19: Table 7. When it is indicated in the table that the sensor network is not used, what it means is that no recommendations are made.

Response: Many thanks to the Reviewer. We have added the following tables and figures to complement the discussion, delving into comparative analyses showing an analytical contribution at the current literature review level.

Comment 20: Table 7. How many recommendations have been made?

Response comments 18, 19, 20: The reviewer is correct; the table was confusing. We have restructured the format of the table, clearly indicating the phases in which the metrics were obtained.

Comment 21: Section 4.2.3 I don't see much use in it. Knowing who has benefited and who hasn't would be enough.

Response: The reviewer is correct, we have dropped all unnecessary description information about clusters and we have added a paragraph where we explain the objective of classifying the samples according to the impact on their metrics through the use of the sensor network.

Labeling samples based on changes in metric performance is essential to perform regression analysis and discover predictive patterns in using the sensor network. Additionally, it allows analysis of differences between metrics before and after the use of the sensor network, for each group of samples isolated by label.

Comment 22: Section 5. This section explains relatively well how the system could be used and how new lines of action could be incorporated. It should be shortened.

Response: Many thanks to the Reviewer. We have shortened the section by including only the key discussions.

Thank you very much.

Sincerely,

Carolina Del-Valle-Soto

Universidad Panamericana. Facultad de Ingeniería. Álvaro del Portillo 49, Zapopan, Jalisco, 45010, México.

Phone: +52 (33) 13682200 | Ext. 4866

Round 2

Reviewer 1 Report

Comments and Suggestions for Authors

The authors have satisfactorily addressed the main concerns.

Reviewer 3 Report

Comments and Suggestions for Authors

The authors have made the requested changes.